



# COSMO-BEP-Tree v1.0: a coupled urban climate model with explicit representation of street trees

Gianluca Mussetti[1,2,3], Dominik Brunner[1], Stephan Henne[1], Jonas Allegrini[2,3], E. Scott Krayenhoff[4], Sebastian Schubert[5], Christian Feigenwinter[6], Roland Vogt[6], Andreas Wicki[6], and Jan Carmeliet[3]

[1]Laboratory for Air Pollution/Environmental Technology, Empa, Dübendorf 8600, Switzerland
[2]Laboratory of Multiscale Studies in Building Physics, Empa, Dübendorf 8600, Switzerland
[3]Chair of Building Physics, ETH Zürich, Zürich 8093, Switzerland
[4]School of Environmental Sciences, University of Guelph, Guelph ON N1G 2W1, Canada
[5]Geography Department, Humboldt-Universität zu Berlin, Berlin 10099, Germany
[6]Research Group Meteorology, Climatology, and Remote Sensing, University of Basel, Basel 4055, Switzerland

**Correspondence:** Dominik Brunner (dominik.brunner@empa.ch) and Gianluca Mussetti (mussetti.gianluca@gmail.com)

**Abstract.** Street trees are more and more regarded as an effective measure to reduce excessive heat in urban areas. However, the vast majority of mesoscale urban climate models do not represent street trees in an explicit manner and for example do not take the important effect of shading by trees into account. In addition, urban canopy models that take interactions of trees and urban fabrics directly into account are usually limited to the street or neighbourhood scale and, hence, cannot be used

to analyse the citywide effect of urban greening. In order to represent the interactions between street trees, urban elements and the atmosphere in realistic regional weather and climate simulations, we coupled the vegetated urban canopy model BEP-Tree and the mesoscale weather and climate model COSMO. The performance and applicability of the coupled model, named COSMO-BEP-Tree, are demonstrated over the urban area of Basel, Switzerland, during the heatwave event of June-July 2015. Overall, the model compared well with measurements of individual components of the surface energy balance and with air and

surface temperatures obtained from a flux tower, surface stations and satellites. Deficiencies were identified for night-time air temperature and humidity, which can mainly be traced back to limitations in the simulation of the night-time stable boundary layer in COSMO. The representation of street trees in the coupled model generally improved the agreement with observations. Street trees produced large changes in simulated sensible and latent heat flux, and wind speed. Within the canopy layer, the presence of street trees resulted in a slight reduction in daytime air temperature and a very minor increase in night-time air

temperature. The model was found to realistically respond to changes in the parameters defining the street trees: leaf area density and stomatal conductance. Overall, COSMO-BEP-Tree demonstrated the potential of (a) enabling city-wide studies on the cooling potential of street trees and (b) further enhancing the modelling capabilities and performance in urban climate modelling studies.

## 1 Introduction

Street trees - trees located in street canyons - and urban vegetation in general are more and more regarded as an effective measure to reduce excessive heat conditions in urban areas (e.g. Shashua-Bar et al., 2009; Armson et al., 2012; Coutts et al.,





2016; Tan et al., 2016; Gunawardena et al., 2017; Manickathan et al., 2018). Excessive heat conditions are typically caused by the combination of regional scale hot weather (heat wave) situations and the urban heat island effect (UHI). The UHI refers to air temperature increase in urban areas compared to their rural surroundings and it is caused by the alteration of the surface energy balance due to the presence of man-made structures and activities (Oke et al., 2017). Climate change is expected to

further accentuate the magnitude and frequency of excessive heat conditions in cities with potentially severe impacts on human health, energy consumption, air pollution and urban ecology (Rosenzweig et al., 2018).

Street trees impact the urban atmosphere through radiative, heat and flow interactions (Oke, 1989). Street trees intercept radiation through their foliage canopy. By intercepting the direct short-wave (solar) radiation, they reduce the temperature of underlying surfaces (Armson et al., 2012). Additionally, street trees also intercept radiation coming from the canyon surfaces,

such as the long-wave radiation emitted by warm wall and street surfaces. At night, this effect can actually reduce the radiative cooling of wall and street surfaces (Bowler et al., 2010). In terms of heat interactions, tree transpiration reduces the surface temperature of the foliage by converting part of the solar radiation to latent heat (Green, 1993). Finally, the tree foliage excerpts momentum from the mean flow, contributing to Turbulent Kinetic Energy (TKE) generation due to wake production and TKE dissipation due to the small scale of the leaves (Wilson and Shaw, 1977).

Modelling studies on the cooling potential of street trees have almost exclusively been performed at the scale of individual street canyons to a single neighbourhood (e.g. Gromke et al., 2015; Ng et al., 2012). Only very few studies investigated the city-wide impact of street trees, mainly due to the limited availability of models able to represent street trees at the scale of the city or at even larger scales. In fact, the vast majority of meso-scale urban climate models only represent vegetation outside the street canyon, neglecting important effects such as the shading effect of trees on the canyon's surfaces.

The very first example of a weather and climate model with an integrated representation of street trees was developed by Dupont et al. (2004). They built an urban canopy model, called DA-SM2-U, for the Penn State–NCAR fifth-generation Mesoscale Model (MM5, Grell et al. (1994)). DA-SM2-U was based on the multi-layer urban canopy concept where a drag-force approach is used to represent the dynamic and turbulent effects of buildings and vegetation. In the meantime, the Weather and Forecasting Model (WRF, Skamarock et al. (2005)) has replaced MM5 and, to the authors best knowledge, DA-SM2-U

was not ported to WRF. Lee and Park (2008) developed an urban canopy model with an explicit representation of street trees named Vegetated Urban Canopy Model (VUCM). VUCM adopts a single-layer urban canopy concept (i.e., the urban canopy is represented by only one model layer) where street trees are represented as a single tree canopy characterised by canopy cover fraction, mean height and mean leaf area density. Lee (2011) further developed VUCM by including a grass-covered soil surface within the canyon. Lee et al. (2016) incorporated VUCM into WRF and performed an application over Seoul, South Korea.

VUCM was also used by Loughner et al. (2012), coupled with WRF, to simulate the role of street trees in Washington, D.C., USA. Wang et al. (2018) incorporated a representation of the radiative effects of street trees (Wang, 2014) into the single-layer urban canopy model of WRF (Chen et al., 2011). They simulated the impact of street trees over the contiguous United States using a 20 km model resolution. Krayenhoff et al. (2018) used WRF, coupled with the single-layer urban canopy model, to simulate the impact of urban adaptation strategies (including street trees planting) for air temperature during contemporary and

future climate over the contiguous United States. A couple of other urban canopy models (UCMs) with explicit representation





of street trees have been developed (Ryu et al., 2016; Redon et al., 2017) but, to the authors best knowledge, not yet coupled with a mesoscale weather and climate model.

The literature review revealed a number of major limitations in the models used to study the city-wide impact of street trees. Several studies neglected some the interactions between street trees and urban fabrics. For example, the exchange of long-wave

radiation between street trees and urban fabrics is often neglected (Wang et al., 2018; Krayenhoff et al., 2018), together with the impact of street trees on the flow field. Neglecting long-wave radiation exchanges and impacts of trees on the flow field may overestimate their cooling potential. Another simplistic assumption regarded the representation of tree shading, which was in some cases independent of the solar direction (Dupont et al., 2004; Lee and Park, 2008).

A second common assumption regards the characterisation of street trees in model applications. In all the previous studies, no

real data on the location, density height and species of street trees were used. Instead, a homogeneous distribution of street trees has often been assumed (Dupont et al., 2004; Loughner et al., 2012). The increasing availability of detailed street trees datasets (e.g. Alonzo et al., 2014; Konarska et al., 2016) gives now a chance to represent the distribution of street trees characteristics more realistically during model applications.

Finally, the majority of the studies did not include a comprehensive evaluation of their model. Typically, only the underlying

urban canopy model used in the coupled model has been evaluated off-line. Online model evaluation against surface stations, flux towers and satellite observations, in the presence of street trees, are needed to better understand the model performance and to identify future lines of development.

Motivated by the great potential of using an integrated model to study the urban climate impact of street trees in a city-wide manner, the objective of this study is two-fold. First, we aim to document the development of an urban climate model

with explicit representation of street trees, featuring a two-way coupling between the vegetated urban canopy model BEP-Tree (Krayenhoff, 2014; Krayenhoff et al., 2019) and the mesoscale weather and climate model COSMO-CLM (COSMO here-inafter; Rockel et al. (2008)). The coupled model, named COSMO-BEP-Tree, represents the multiple interactions between street trees, urban fabrics and the atmosphere in a comprehensive way considering the exchange of heat, moisture, and momentum and the transfer of short- and long-wave radiation. COSMO-BEP-Tree enables city-wide studies on the cooling potential

of street trees in comparison, or in combination, with other UHI mitigation measures (e.g. albedo changes). Additionally, it has the potential to enhance the modelling performance in urban climate modelling studies. In order to facilitate the application, the model comes with a tool to pre-process the additional inputs required for COSMO-BEP-Tree, such as street trees datasets or a 3D city model.

Second, we present an application and comprehensive evaluation of COSMO-BEP-Tree over the transnational urban ag-

glomeration of Basel (Switzerland, Germany and France) during a heatwave event in June-July 2015. The evaluation makes use of the extensive measurement infrastructure for urban climate studies available in Basel (Feigenwinter et al., 2018; Wicki et al., 2018) including an urban flux tower and a network of surface stations and additionally uses land surface temperature (LST) observations from satellite. In order to verify the model's response to the parameters that define the street trees, a sensitivity analysis is also presented.





We remark that this study only aims to document the development and evaluation of COSMO-BEP-Tree. Analysis on the role of street trees in urban climate and on their potential for mitigating excessive heat conditions are out of the scope of this paper and will be presented in follow-up studies. Although the focus of this study is on street trees, it is important to consider that other components of the so-called urban forest, such as trees in parks and private garden as well as peri-urban trees, are

also likely to have a remarkable impact on the urban climate.

The manuscript is structured as follows. The two model components COSMO and BEP-Tree, together with the coupling strategy and the pre-processor, are described in Section 2. Section 3 presents the model evaluation and sensitivity. In Sections 5 and 6 future work and conclusions are discussed.

## 2   Model description

### 2.1   COSMO model

The Consortium for Small-scale Modelling (COSMO) model is a non-hydrostatic limited-area atmospheric prediction model (COSMO). It has been designed for operational numerical weather prediction, regional climate simulation and other applications at the mesoscale (weather phenomena of sizes between about 1 km and a few hundred kilometres). COSMO evolved from the operational weather forecast Local Model (LM) of the German Weather Service (Steppeler et al., 2003) and has

been developed by a consortium of weather services in Europe and by the CLM-Community for climate applications (CCLM-Community). The regional climate model version of COSMO, called COSMO-CLM, includes modifications allowing the application on time-scales up to centuries (Rockel et al., 2008). These modifications comprise a representation of phenological cycles as well as externally prescribed, time-dependent atmospheric $CO_2$ concentrations.

The COSMO model is based on the thermo-hydrodynamical equations describing non-hydrostatic compressible flow in a

moist atmosphere. A variety of physical processes are taken into account by parameterisation schemes: sub-grid scale turbulence, surface fluxes, grid and sub-grid scale clouds and precipitation, moist and shallow convection, radiation, soil and vegetation, lake and sea ice schemes (see Figure 1a). At the convection-resolving model resolution used in this study, the parameterisations for sub-grid scale clouds and moist convection are switched off. Additional information on the governing equations and physical parameterisations can be found in the model documentation (COSMO).

Recently, urban parameterisations of different degrees of complexity have been developed for and coupled with the COSMO model (Trusilova et al., 2016). The urban parameterisation TERRA-URB (Wouters et al., 2015, 2016) employs a simple but efficient approach by representing the urban canopy using a limited number of parameters (so-called bulk approach). The Town Energy Balance (TEB) implementation of Trusilova et al. (2013) represents an intermediate level of complexity with a single-layer urban canopy model. Finally, the multi-layer urban canopy model DCEP (Double-Canyon Effect parameterisation)

employs the most physically-based but input-demanding representation of the urban canopy by solving the radiation exchange in a realistic multi-layer canyon configuration (Schubert et al., 2012).



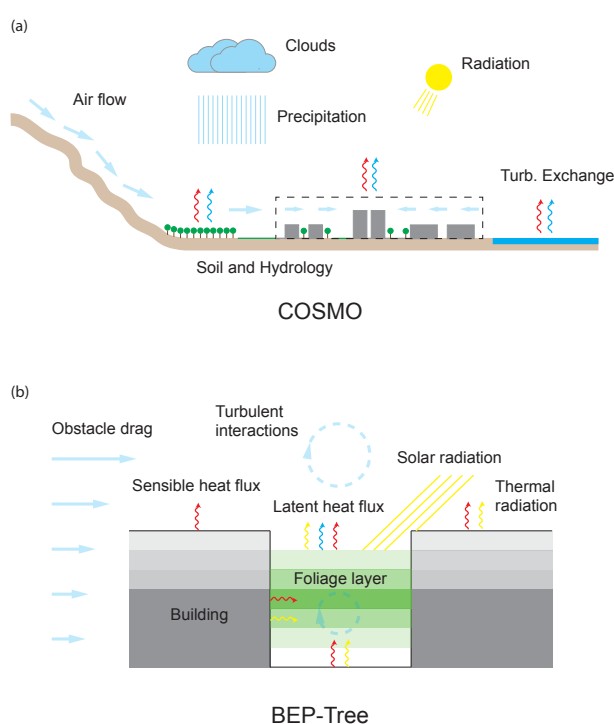

**Figure 1.** Schematic of the key processes represented by (a) COSMO and (b) BEP-Tree for urban climate simulations. Details about the interactions between radiation and the foliage layers (e.g. radiation absorption and scattering) are not represented here. Please refer to the Sections 2.1 and 2.2 for a comprehensive description of the models.

Applications of these models included studies of the impact of climate change on cities (Grossman-Clarke et al., 2017; Wouters et al., 2017), the influence of green areas and low albedo materials (Schubert and Grossman-Clarke, 2013) and the intra-urban climate variability (Mussetti et al., 2019).

## 2.2 Building Effect parameterisation with Trees (BEP-Tree)

5 The Building Effect Parameterisation with Trees (BEP-Tree) is a multi-layer urban canopy model with explicit representation of trees and their interaction with the street canyon (Krayenhoff, 2014; Krayenhoff et al., 2019). Building on top of the geometry of the existing multi-layer urban canopy model BEP (Martilli et al., 2002), Krayenhoff et al. (2014) included the effects of tree foliage on the radiative exchange within the street canyon.

The basic geometry is a two-dimensional canyon with vertical leaf area density profiles and probabilistic variation of building
10 height (see Figure 1b). Tree foliage is permitted both between and above buildings allowing the representation of street trees as well as trees overhanging buildings.

Ray tracing is used to determine the receipt of direct short-wave irradiance by building and foliage elements. View factors for long-wave and short-wave diffuse radiation exchange are computed once at the start of the simulation using a Monte Carlo





ray tracing approach, and used thereafter to calculate multiple reflections between all urban fabrics. Interception of radiation by tree foliage is modelled with the Bouguer-Lambert-Beer law:

$$\Delta V_i = r_i \Big( 1 - \exp(-K\,\Omega\,L_D\,\Delta s\,f_i) \Big), \tag{1}$$

where $\Delta V_i$ is the reduction in intensity of ray $i$ due to the tree foliage [-], $r_i$ is the initial intensity of the ray $i$ [-], $K$ is the
foliage extinction coefficient [-], $L_D$ is the leaf area density [m$^2$ m$^{-3}$], $\Omega$ is the foliage clumping index [-], $\Delta s$ is the 2-D ray step size [m] and $f_i$ is the ratio of 3-D (actual) to 2-D (model) distance travelled by the ray (actual 3-D paths of all rays are mapped to the 2-D domain, depending on the canyon orientation; Krayenhoff et al. (2014)).

BEP-Tree includes a parameterisation of building and tree foliage effects on the airflow (Martilli et al., 2002; Santiago and Martilli, 2010; Krayenhoff et al., 2015). The parameterisation introduces (a) a sink term for momentum to account for obstacle
drag, (b) modifications of turbulence length scales by building interactions, and (c) enhanced dissipation of turbulent kinetic energy due to the small scale of the tree foliage elements. The obstacle drag from tree foliage is determined as:

$$S_{m,i} = -C_{dv}\,\Omega\,L_D\,U\,\overline{u_i}, \tag{2}$$

where $S_{m,i}$ is the sink term for momentum along the coordinate axis $i$ [m s$^{-2}$], $C_{dv}$ is the sectional drag coefficient for tree foliage (set to 0.2) [-], $U$ is the horizontal wind speed [m s$^{-1}$] and $\overline{u_i}$ is the wind speed component along axis $i$ [m s$^{-1}$].

Surface energy balance equations for roof, wall and road elements are identical to Martilli et al. (2002). For leaf layers, the Penman equation is rearranged to solve for the leaf temperature as in Campbell and Norman (2012). The sensible heat flux, latent heat flux and source of moisture from street trees are determined as:

$$Q_H = c_p\,g_{Ha}\,\Omega\,L_D\,(T_{\text{veg}} - T_{\text{air}}), \tag{3}$$

$$Q_E = \lambda\,\frac{g_s g_a}{g_s + g_a}\,\Omega\,L_D\Big(s\,(T_{\text{veg}} - T_{\text{air}}) + \frac{e_s - e_{\text{air}}}{p}\Big), \tag{4}$$

$$S_q = \frac{\mu_w}{\rho_{\text{air}}}\,\frac{g_s g_a}{g_s + g_a}\,\Omega\,L_D\Big(s\,(T_{\text{veg}} - T_{\text{air}}) + \frac{e_s - e_{\text{air}}}{p}\Big), \tag{5}$$

where $Q_H$ is the sensible heat flux from foliage per unit height [W m$^{-3}$], $c_p$ is the specific heat capacity of air [J mol$^{-1}$ K$^{-1}$], $g_{Ha}$ is the heat conductance [mol m$^{-2}$ s$^{-1}$], $T_{\text{veg}}$ is the temperature of the foliage layer [K], $T_{\text{air}}$ is the air temperature at the height of the foliage layer [K], $Q_E$ is the latent heat flux from foliage per unit height [W m$^{-3}$], $\lambda$ is the latent heat of vaporisation [J mol$^{-1}$], $g_s$ is the vapour stomatal conductance of the leaf [mol m$^{-2}$ s$^{-1}$], $g_a$ is the vapour boundary-layer
conductance of the leaf [mol m$^{-2}$ s$^{-1}$], $\rho_{\text{air}}$ is the density of air [kg m$^{-3}$], $s = \frac{de_s}{dT_{\text{air}}}\frac{1}{p}$ is the change is saturation pressure with temperature given by the Clausius-Clapeyron equation (Stull, 2012) [K$^{-1}$], $e_s$ is the saturation vapour pressure [Pa], $e_{\text{air}}$ is the vapour pressure in the air [Pa], $p$ is the atmospheric pressure [Pa], $S_q$ is the source term for moisture due to transpiration from foliage [s$^{-1}$] and $\mu_w$ is the molar mass of water [kg mol$^{-1}$]. Here we report only those equations that are relevant for the coupling of BEP-Tree with COSMO and for understanding the response to the sensitivity experiments described later. The
complete set of equations as well as further details about the model components can be found in Krayenhoff et al. (2014, 2015, 2019), including a recently-developed parameterisation for climate impacts of street tree foliage distribution.





The main model inputs to BEP-Tree are (a) atmospheric state variables above the canyon and (b) canyon geometry including vegetation. The former includes the traditional atmospheric variables and radiation fluxes, that can be either provided by a tall tower (off-line application) or by a weather model (online application). The latter includes height distribution of buildings, canyon width, thermo-physical material properties of active surfaces (wall elements, roof and street) and parameters representing in-canyon vegetation.

Regarding in-canyon vegetation, a vertical profile of leaf area density ($L_D$) needs to be provided for both the canyon and the building columns. $L_{D,can}$ and $L_{D,bld}$ represent the surface area of leaves per unit volume of air [$m^2\ m^{-3}$] in the canyon space and in the building space (e.g. in case of trees overhanging buildings), respectively. Additionally, information on the spatial distribution of the leaves from street trees in the available canyon space needs to be provided. This information is represented by the clumping index $\Omega$ as defined by Nilson (1971). The index takes values between 0 and 1, with $\Omega = 1$ being the case where the leaves are randomly distributed. $\Omega$ encompasses clumping at several scales, from the within-branch to the between-crown scale. $\Omega$ is used, together with $L_D$, to account for the fraction of leaves that actively contribute to the radiation exchange. Moreover, $\Omega$ is used to represent the aerodynamic sheltering following the approach proposed by Marcolla et al. (2003) and to modulate wake production and turbulent kinetic energy (TKE) dissipation (Krayenhoff, 2014). $\Omega$ is always used in combination with $L_D$ to compute an "effective leaf area density" defined as $L_{De} = L_D\,\Omega$.

BEP-Tree has been evaluated in off-line mode against a measurement site in Vancouver, Canada (Krayenhoff, 2014). The evaluation made use of measured fluxes of radiation and turbulent exchange in the inertial sublayer, as well as measurements of air temperature and humidity in the urban canopy layer. The evaluation has been recently extended using measurements of street surface temperature and street-level radiation fluxes from two sites in North America (Krayenhoff et al., 2019).

## 2.3 COSMO-BEP-Tree

We coupled BEP-Tree to the COSMO-CLM model v5.0_clm2.1. The coupling is done through model integration (also called joint coupling by Brandmeyer and Karimi (2000)) and makes use of the interface created by Schubert et al. (2012) for the urban parameterisation DCEP. The coupled model, named COSMO-BEP-Tree, is developed to simulate the exchange of energy, momentum, and moisture between an urban canopy with street trees and the atmosphere and the effects of buildings and trees on TKE during weather and climate simulations.

The coupling makes use of the tile approach, where mixed configurations of urban and natural surfaces within a grid cell can be considered. Natural surfaces within a urban grid cell represents, for instance, urban parks and peri-urban forests. The surface-atmosphere interactions are calculated independently for the urban and rural tiles and then averaged with respect to the fractional coverage of each tile. BEP-Tree is used to calculate the surface-atmosphere interactions for the urban tile, while the land surface model of COSMO does it for the natural tile (see Section 2.1 for further details).

A schematic representation of the coupling is presented in Fig. 2. Before executing COSMO-BEP-Tree, the parameters that characterise the street canyon are estimated for all urban grid cells in the model domain from external input datasets of trees, building geometries and imperviousness. This task is performed by a dedicated pre-processor described in Sec. 2.4. Once all required inputs are generated, the coupled model can be executed. At the first model time step, the urban canopy parameters





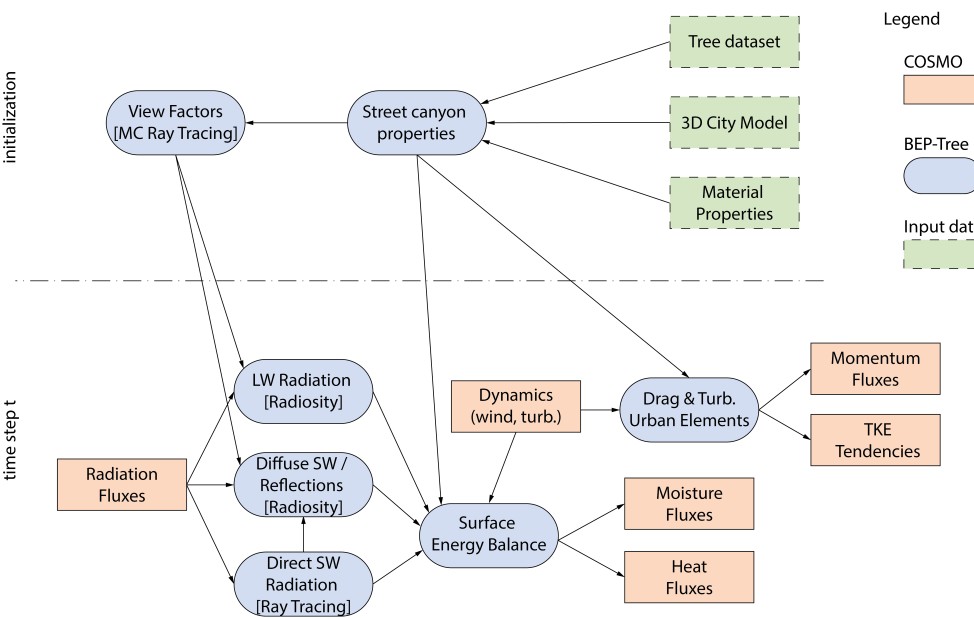

**Figure 2.** Flow chart of the coupling of BEP-Tree with COSMO. Rose boxes denote operations of COSMO, blue boxes of BEP-Tree. Additional input datasets required to run COSMO-BEP-Tree are shown as green boxes.

are used to estimate the view factors for long-wave and diffuse short-wave radiation between each street canyon element (sky, wall, roof, street and foliage layer). The view factors are stored in the form of a matrix and applied during the model simulation at each time step. In the case of direct short-wave radiation, the receipt of radiation by each street canyon element is computed online during the simulation using a direct ray tracing algorithm. Subsequently, the diffuse short-wave radiation, short-wave

reflections and received and emitted long-wave radiation at each street canyon element are estimated using a radiosity approach (making use of the view factor matrices). At this point, the radiation balance at each active surface is available and is used to determine the radiation balance of the entire grid cell in the COSMO model.

The user can specify the frequency at which the radiation exchanges in BEP-Tree are updated. By default, this is set to 0.05 h (3 minutes), as a compromise between accuracy and computational cost.

The air temperature tendencies (increments) due to sensible heat flux, the momentum tendencies due to obstacle drag and the TKE tendencies are determined for each street canyon element. For the foliage layers, latent heat and moisture fluxes are calculated by solving the leaf energy balance. Although moisture exchange between street trees and the atmosphere is implemented, street trees do not interact with soil moisture content as represented by COSMO's land surface scheme. In other words, a mechanistic interaction between soil moisture and the transpiration of street trees is not included, assuming that the

transpiration is never limited by soil water availability.




The radiative coupling between COSMO and BEP-Tree is realised through four bulk radiation parameters: (a) albedo for diffuse short-wave radiation $\alpha^{\downarrow}_{\mathrm{urb}}$, (b) albedo for direct short-wave radiation $\alpha^{\Downarrow}_{\mathrm{urb}}$, (c) mean emissivity $\epsilon_{\mathrm{urb}}$ and (d) radiative surface temperature $T_{\mathrm{urb}}$. $\alpha^{\downarrow}_{\mathrm{urb}}$ does not change with time being only dependent on the canyon geometry and on the albedo of individual surface elements, given the assumption of isotropic diffuse radiation exchange. Therefore, it is calculated only once

at the beginning of the simulation as:

$$\alpha^{\downarrow}_{\mathrm{urb}} = \frac{K^{\uparrow_0}_{\mathrm{urb}}}{K^{\downarrow}}, \tag{6}$$

with $K^{\uparrow_0}_{\mathrm{urb}}$ being the shortwave radiation reflected from all street canyon elements into the sky for the case where the incoming direct short-wave radiation $K^{\Downarrow}$ is zero and $K^{\downarrow}$ being the incoming diffuse short-wave radiation.

$\alpha^{\Downarrow}_{\mathrm{urb}}$, instead, depends on the position of the sun and is therefore estimated at each time step as:

$$\alpha^{\Downarrow}_{\mathrm{urb}} = \frac{(K^{\uparrow}_{\mathrm{urb}} - \alpha^{\downarrow}_{\mathrm{urb}} K^{\downarrow})}{K^{\Downarrow}}, \tag{7}$$

with $K^{\uparrow}_{\mathrm{urb}}$ being the shortwave radiation reflected from all street canyon elements into the sky.

The radiative surface temperature is calculated as:

$$T_{\mathrm{rad}} = \left( \frac{L^{\uparrow}_{\mathrm{urb}} - (1 - \epsilon_{\mathrm{urb}}) L^{\downarrow}}{\sigma \, \epsilon_{\mathrm{urb}}} \right)^{1/4}, \tag{8}$$

where $L^{\uparrow}_{\mathrm{urb}}$ is the sum of the emitted and reflected long-wave radiation from all street canyon elements into the sky, $\epsilon_{\mathrm{urb}}$ is the

bulk emissivity of the urban surface, $L^{\downarrow}$ is the incoming long-wave radiation and $\sigma$ is the Stephan-Boltzmann constant. $\epsilon_{\mathrm{urb}}$ is calculated as the mean emissivity of the street canyon elements.

The bulk radiation parameters from the urban tile ($_{\mathrm{urb}}$) computed by BEP-Tree are combined with the ones computed by COSMO for the natural ($_{\mathrm{nat}}$) tile as:

$$\alpha^{\Downarrow} = \mathrm{f}_{\mathrm{urb}} \, \alpha^{\Downarrow}_{\mathrm{urb}} + \mathrm{f}_{\mathrm{nat}} \, \alpha^{\Downarrow}_{\mathrm{nat}}, \tag{9}$$

$$\alpha^{\downarrow} = \mathrm{f}_{\mathrm{urb}} \, \alpha^{\downarrow}_{\mathrm{urb}} + \mathrm{f}_{\mathrm{nat}} \, \alpha^{\downarrow}_{\mathrm{nat}}, \tag{10}$$

$$\epsilon = \mathrm{f}_{\mathrm{urb}} \, \epsilon_{\mathrm{urb}} + \mathrm{f}_{\mathrm{nat}} \, \epsilon_{\mathrm{nat}}, \tag{11}$$

$$T_{\mathrm{rad}} = \left( \frac{(\mathrm{f}_{\mathrm{urb}} \, \epsilon_{\mathrm{urb}} \, T^4_{\mathrm{urb}} + \mathrm{f}_{\mathrm{nat}} \, \epsilon_{\mathrm{nat}} \, T^4_{\mathrm{nat}})}{\epsilon} \right)^{1/4}, \tag{12}$$

with $\mathrm{f}_{\mathrm{urb}}$ and $\mathrm{f}_{\mathrm{nat}}$ (= 1 - $\mathrm{f}_{\mathrm{urb}}$) being the planar fractions of urban and natural surfaces, respectively. The resulting bulk radiation parameters are used as lower (surface) boundary conditions for the short-wave and long-wave radiation routines in COSMO.

In the same way, the tendencies of momentum, sensible heat, moisture and TKE production from the urban tile are combined with the corresponding tendencies from the natural tile.

BEP-Tree operates on a grid (called urban grid) that is vertically independent of and generally finer than the grid of COSMO (called mesoscale grid). As a default, the vertical resolution of the urban grid is 5 m. At each model time step, the input fields





from COSMO have to be interpolated to the urban grid whereas the output fields (tendencies) of BEP-Tree have to be aggregated back to the mesoscale grid. The interpolation and aggregation of fields in COSMO-BEP-Tree follows the implementation of Schubert (2013).

In analogy to what done by Schubert (2013), the 2 m air temperature is diagnosed from the air temperature at the lowest
mesoscale model layer and the surface temperature from both urban and natural surfaces as:

$$T_{2m} = (f_{nat}\, T_g + f_{urb}\, T_s) + r\, (T_1 - (f_{nat}\, T_s + f_{urb}\, T_g)), \tag{13}$$

where $T_g$ is the surface temperature of the natural tile, $T_s$ is the surface temperature of the urban tile, $r$ is a coefficient that represents the stability-dependent resistance for scalar transport as estimated by the TKE-based surface-layer transfer scheme of COSMO (Baldauf et al., 2011), and $T_1$ is the air temperature at the lowest model level.

**2.4   COSMO-BEP-Tree preprocessor**

Additional input variables need to be calculated before running COSMO-BEP-Tree. These variables, referred to as urban canopy parameters (UCPs) in the following, describe the spatial representation of the street canyon (including street trees) required by the urban canopy model. A dedicated python program, UPCgenerator v1.0, has been developed to generate the UCPs based on external datasets. The workflow of UCPgenerator is shown in Fig. 2 (green boxes). A technical description of
UCPgenerator v1.0 is given in Appendix A.

The UCPgenerator requires the following three input datasets: (1) street trees, (2) buildings and (3) imperviousness. The dataset of street trees consists of a spatial map of the street tree distribution. This may include information on tree height, leaf area index ($L$) or leaf area density ($L_D$) and tree species. In the case of missing information, representative profiles or knowledge-based assumptions can be used instead. Information on street trees is typically provided by inventories maintained
by urban environmental agencies (e.g. Stadtgaertnerei Kantons Basel-Stadt), remotely sensed data such as LIDAR data (e.g., Alonzo et al., 2014) and combined data derived using data fusion techniques (e.g., Branson et al., 2018).

The building dataset consists of a collection of building geometries in the urban area, including building height as an attribute of each individual building. 3D building datasets are typically provided by national mapping agencies and collaborative projects (e.g. OpenStreetMap).

The imperviousness dataset is a spatial representation of the horizontal fraction of impervious surfaces in the model domain. Such data is available at the European scale (e.g. EEA, 2015) as well as at the global scale (e.g. Brown de Colstoun and Wolfe, 2017).

**3   Model evaluation**

**3.1   Study area and model set up**

COSMO-BEP-Tree was applied over the Basel metropolitan area (Basel hereinafter), located mainly in Switzerland but extending over the borders of France and Germany (Fig. 3). Situated along the Rhine river, the larger Basel area has a population





of approx. 830'000 inhabitants and is surrounded by hilly to mountainous terrain especially towards the south (Jura mountains) and north-east (Black Forest). According to the Köppen climate classification, Basel features a temperate oceanic climate (Köppen: Cfb). The inner city (Basel-Stadt) is characterised by an abundance of green areas with more than 24'000 urban trees and 275 hectares of public vegetated surfaces distributed over a total area of the city centre of approx. 2'385 hectars (Stadt-

gaertnerei Kantons Basel-Stadt). The city of Basel is not only an interesting target for its abundant green infrastructures, but also for the availability of extensive observational data sets for model evaluation, which had been used in numerous previous urban climate studies (e.g. Rotach et al., 2005; Parlow et al., 2014). Observations and input data used here are described in Section 3.3 and Section 3.2.

   In this study, COSMO-BEP-Tree was run for the period 22 June 2015 - 9 July 2015, with the first 5 days discarded as spin-

up period. The period corresponds to an intense heatwave that persisted across Europe producing a seasonal mean surface air temperature 2.4 K above the 1964-1993 mean (Dong et al., 2016; Ionita et al., 2017). The event featured a stable anticyclonic weather regime with a prevalence of cloud-free conditions, low wind speeds and absence of precipitation. The conditions were ideal for evaluating the model performance, particularly with respect to the impact of street trees on the urban atmosphere and the development of the UHI.

We applied COSMO-BEP-Tree over a domain of approx. 90 km x 80 km with a horizontal grid spacing of 0.0025° (approx. 270 m). 60 levels were used in the vertical direction, with 5 and 23 levels in the first 100 m and 1000 m, respectively. The model set-up closely follows the configuration described in Mussetti et al. (2019). Initial and boundary conditions were taken from the operational COSMO-2 analyses operated by the Swiss Federal Office of Meteorology and Climatology (MeteoSwiss), which are centred over Switzerland and cover the whole Alpine domain. The analyses have a horizontal resolution of about 2 km and

are produced using a nudging technique (Schraff, 1997) applied to near-surface observations and vertical soundings of pressure, humidity and wind. As additional input parameters for COSMO, the 1" global digital elevation map ASTER (Tachikawa et al., 2011) for topography, the 10" land use dataset GlobCover 2009 (Loveland et al., 2000) for land cover, and the 30" Harmonized World Soil Database (FAO et al., 2009) for soil properties were used.

   To assess the physical consistency of the implementation of the different interactions with street trees, we performed ad-

ditional sensitivity simulations changing the values of the parameters describing the street trees (Table 1). In addition to the standard run (STD), simulations without street trees (LA0), with increased and decreased values of leaf area density (LA+ and LA-, respectively), and with increased and decreased values of stomatal conductance (SC+ and SC-, respectively) were performed.

### 3.2   Urban canopy input data

Figure 4 shows the distribution of the main urban canopy parameters as derived by the preprocessor UCPgenerator. The model represents the spatial variability of the urban texture in terms of fraction of urban surfaces, street canyon orientation and width , building width and urban vegetation (Figures 4e-f). $L_{\mathrm{can}}$ and $L_{\mathrm{nat}}$ are obtained by vertical integration of the respective leaf area densities $L_{\mathrm{D,can}}$ and $L_{\mathrm{D,nat}}$. Figure 4 also gives a flavor of the heterogeneity of the city: Industrial areas in the northern and south-east parts of the city, for instance, are associated with wide street canyons and very low vegetation density. Residential



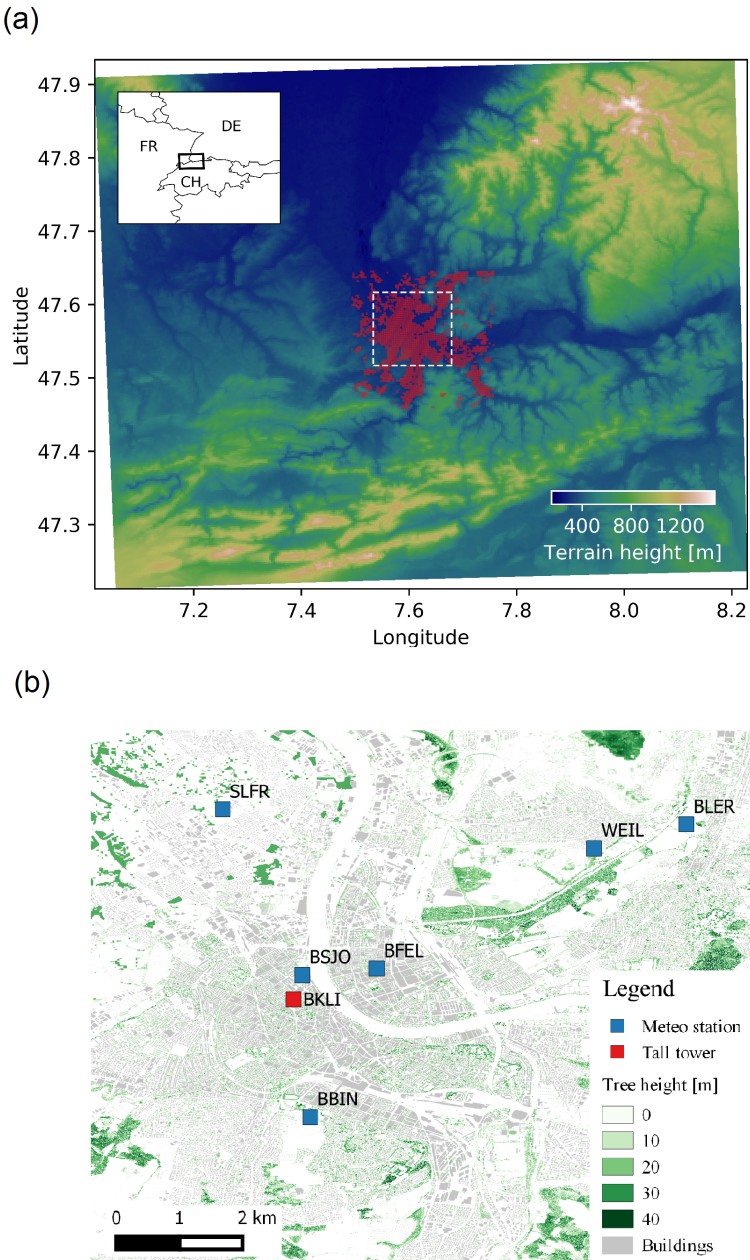

**Figure 3.** (a) The model domain centred over the metropolitan area of Basel. The terrain height (in m a.s.l.) is shown in the background. The urban area is highlighted in red. (b) Detail of the urban area with the location of the measurement sites used for the evaluation (Section 3). In the background are the tree height distribution (green) (Basel-Stadt, 2019) and building geometries (grey).





**Table 1.** Summary of numerical experiments and changes in the parameters used for the sensitivity study. Std indicates the standard value(s) as provided in Section 3.2. The other parameters are kept constant with the values provided in Section 3.2.

| Name | $L_{\mathrm{De}}$ | $g_{\mathrm{s}}$ |
|------|------|------|
| STD | Std | Std |
| LA0 | 0 | Std |
| LA+ | +50% | Std |
| LA- | -50% | Std |
| SC+ | Std | +50% |
| SC- | Std | -50% |

areas in the centre-south and centre-west, in contrast, are characterised by narrow street canyons and relatively high vegetation density.

The urban canopy parameters were derived with UCPgenerator from the following input datasets: (a) imperviousness from EEA (2015), (b) 3D buildings from Federal Office of Topography (2007) and (c) lidar-based tree canopy height (see data

availability section). The tree canopy height dataset has a 2 m spatial resolution and includes all trees in the Basel area. The imperviousness data set was used to differentiate between street trees (contributing to $L_{\mathrm{D,can}}$) and trees over natural surfaces (contributing to $L_{\mathrm{D,nat}}$). Specifically, trees located over mostly impervious surface ($\geq$ 50%) are classified as street trees. A local leaf area density (inside individual trees) $L_{\mathrm{D,loc}}$ of 1 m$^3$ m$^{-2}$ was used according to measurements over urban trees during summertime in Gothenburg, Sweden (Klingberg et al., 2017). The tree species measured by Klingberg et al. (2017) are assumed to be rather representative as they correspond to 6 of the 12 most common species present in Basel (Basel-Stadt,

2019). The density of street trees above the building column ($L_{\mathrm{D,bld}}$, representing trees overhanging roofs) was assumed to be 0. This assumption was made giving that no reliable information was available about the overlapping between tree crowns and roofs. Further details regarding the methodology used to derive the urban canopy parameters are given in the Appendix A. The resulting $L_{\mathrm{can}}$ (Figure 4e) have values ranging from 0 up to 3 m$^2$ m$^{-2}$, which is within the same range as reported in previous

urban studies (Liss et al., 2010; Alonzo et al., 2015; Klingberg et al., 2017).

The thermal and physical properties of urban elements are listed in Table 2. The values are based on the recommendations of Loridan and Grimmond (2012) and are used as default values in COSMO-BEP-Tree. The values have already been evaluated over Basel by Loridan and Grimmond (2012). A foliage scattering coefficient of 0.5 and a foliage emissivity of 0.95 are used according to Krayenhoff (2014).

A step function is used to model the stomatal conductance ($g_s$) with values of 140 mmol m$^{-2}$ s$^{-1}$ and 20 mmol m$^{-2}$ s$^{-1}$ for daytime and night-time, respectively. The values are calculated combining the dataset of urban tree species in the public area of Basel (Basel-Stadt, 2019) with observed values of stomatal conductance (Konarska et al., 2016; Keel et al., 2007; Campbell and Norman, 2012; Xiong et al., 2018). Additional details are provided in the Supplement.



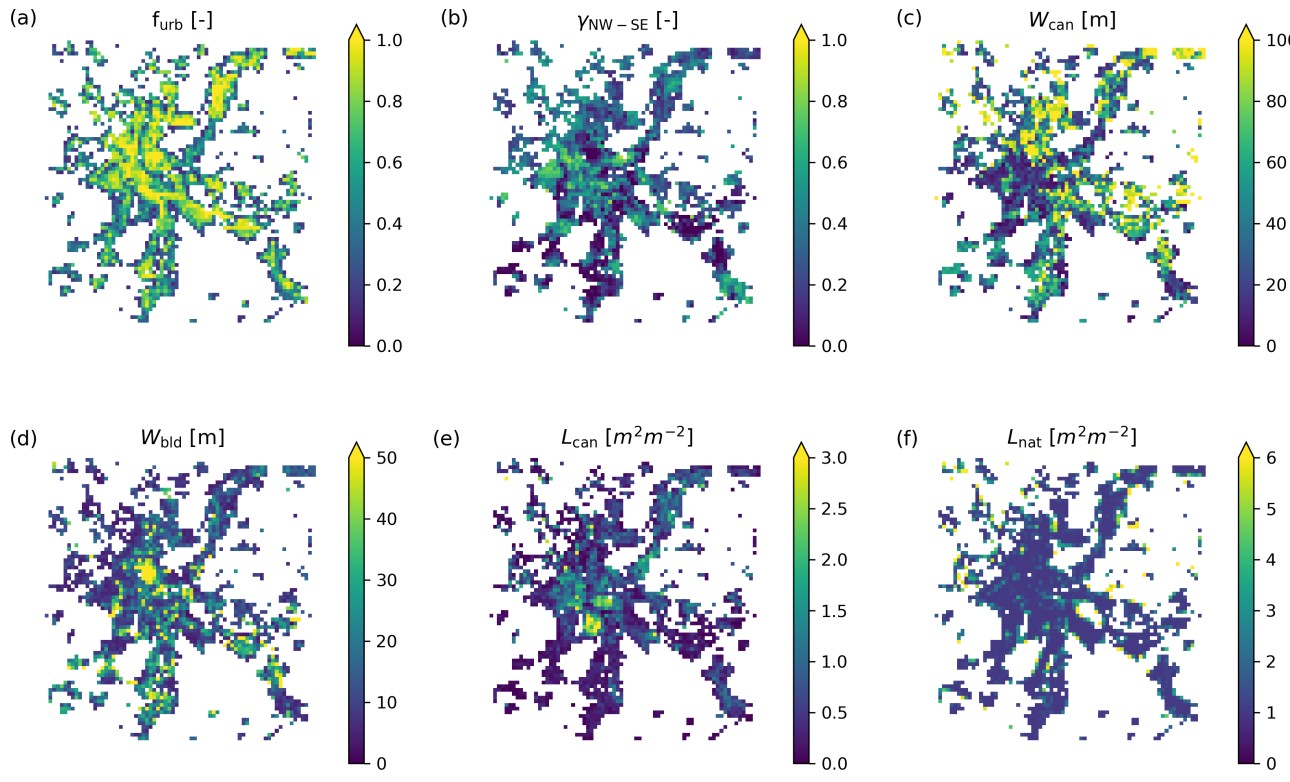

**Figure 4.** Urban canopy parameters derived for the model domain: (a) urban fraction, (b) canyon fraction along the NW-SE direction, (c) average canyon width, (d) average building width, (e) leaf area index due to street trees, and (f) leaf area index due to vegetation outside the urban canopy. Further details about the variables in Table A1. Only urban grid cells ($f_{urb} > 0.1$) are shown.

**Table 2.** Material properties used in the urban canopy model.

|  | Roof | Wall | Road |
|---|---|---|---|
| Albedo (-) | 0.1 | 0.1 | 0.15 |
| Emissivity (-) | 0.85 | 0.9 | 0.95 |
| Heat Capacity (J m$^{-3}$ K$^{-1}$ × 10$^6$) | 1.5 | 1.4 | 1.5 |
| Heat Conductivity (W m$^{-1}$ K$^{-1}$) | 0.8 | 1.0 | 0.8 |
| Total thickness (m) | 0.5 | 0.3 | 1.0 |

A uniform constant value of clumping index ($\Omega$) of 0.5 was used for the reference simulation (STD). In the absence of easily applicable methods to determine $\Omega$ in street canyon environments, we made our choice in analogy to measurements over an open Savannah (Ryu et al., 2010). Savannah is a natural environment that resemble the urban environment in terms of heterogeneity and openness of the tree canopy.





**Table 3.** Overview of observation sites used for model evaluation. Figure 3b shows the location of the sites. $T$ is air temperature, $q$ specific humidity, $u$ wind speed, $Q_H$ sensible heat flux, and $Q_L$ latent heat flux.

| Name | Variable[a] | Type | Height | $f_{urb}$ | $L_{can}$ | LCZ |
|------|----------|------|--------|-----------|-----------|-----|
| BKLI | $T, q, u, Q_H, Q_L$ | Flux tower | 38 m | 0.79 | 1.50 | Compact mid-rise (2) |
| BFEL | $T$ | Surface | 2 m | 0.98 | 1.03 | Compact mid-rise (2) |
| BSJO | $T$ | Surface | 2 m | 0.85 | 1.06 | Compact mid-rise[b] (2) |
| SLFR | $T$ | Surface | 2 m | 0.51 | 0.35 | Large low-rise (8) |
| BBIN | $T$ | Surface | 2 m | <0.1 | 0.00 | Rural low plants[c] (D) |
| BLER | $T, q, Q_H, Q_L$ | Surface | 2 / 10 m | <0.1 | 0.00 | Rural low plants (D) |

[a] Only variables used in this study are reported (the station may measure additional variables). [b] with scattered trees (LCZ B). [c] surrounded by open mid-rise residential areas (LCZ 2).

Anthropogenic heat flux ($Q_A$) was estimated off-line with the Large scale Urban Consumption of energY model (LUCY; Allen et al. (2011); Lindberg et al. (2013)) version 2014a. LUCY simulates the hourly heat emissions from buildings, vehicles and human metabolism using gridded inventories of population density, primary energy consumption, air temperature, number of vehicles, seasonal, weekly, and diurnal cycles of traffic density. Gridded population density from Center for International

Earth Science Information Network (CIESIN) with a resolution of 30" were used. Monthly values of energy consumption in the region of Basel have been obtained for 2015 from Basel-Stadt replacing the default yearly average value, as recommended by Lindberg et al. (2013). The model estimated period-average values of $Q_A$ of 4 and 14.4 W m$^{-2}$ over the entire urban area and the city centre, respectively. A maximum value of 22.9 W m$^{-2}$ was found around mid-day in the city centre. The values are consistent with previous estimates of 20 W m$^{-2}$ as annual average from Christen (2005).

**3.3 In situ measurements**

Measurements from one 38 m tall flux tower and six surface stations were used to evaluate the model results. Table 3 shows a summary of the corresponding observations. The site of Basel Klingelbergstrasse (BKLI) is an urban flux tower operated by the MRC-Lab (Meteorology, Climatology and Remote Sensing) of the University of Basel. BKLI is located in an area with a dense mix of mid-rise building, corresponding to a Local Climate Zone (LCZ) 2 (Stewart and Oke, 2012), and relatively high density

of street trees. The site consists of a 18 m high tower mounted on the flat roof of a 20 m high building. According to Lietzke and Vogt (2013) and Feigenwinter et al. (2012), the measurements can be considered to take place above the roughness sub-layer, but inside the inertial sub-layer. The footprint area for energy fluxes was estimated to 0.60 km$^2$ for 90% of the source area in summer (Schmutz et al., 2016). The site comprises an eddy covariance system consisting of an ultrasonic anemometer (HS-50, Gill Instruments Ltd.) and an open-path $CO_2/H_2O$ infra-red gas analyser (LI-7500, LI-COR Inc.). Air temperature is measured

with a fine-wire thermocouple mounted in a fan-aspirated radiation shield (ASPTC, Campbell Scientific); relative humidity is measured with a mirror-type dew point hygrometer (THYGAN VTP6, Meteolabor AG); radiation fluxes are measured with a net radiometer (CNR4, Kipp & Zonen). It is worth to mention that the flux measurement is characterised by a considerable





uncertainty (Richardson et al., 2012; Järvi et al., 2018). The uncertainty due to random errors is approx. 20% and 30% of the measured flux of sensible and latent heat, respectively (Hollinger and Richardson, 2005). Further details on the site and data processing are available in Schmutz et al. (2016); Lietzke and Vogt (2013); Feigenwinter et al. (2018).

Six surface stations are used to evaluate the model performance inside the urban canopy layer. The sites of Basel Feld-
bergstrasse (BFEL) and Basel St. Johannplatz (BSJO) are operated by the local environmental agency (Lufthygieneamt beider Basel). The site of BFEL is located in an area corresponding to compact mid-rise (LCZ 2) and has moderate density of street trees. It is placed directly inside the street canyon at the intersection between two roads oriented along the 80°/260° and 160°/340° axis. Due to its vicinity to a south facing wall, BFEL may experience temperatures larger then those expected in the area. The site of BSJO is located in a small park at the border of the old city core. The park is vegetated and extends to the
north-east while the surrounding area is characterised by a dense mix of mid-rise buildings. Both sites BFEL and BSJO measure temperature with a thermometer with radiation shield (HC-S3, Campbell Scientific). The site of Saint-Louis Trois-Frontières (SLFR) is operated by the environmental agency of Alsace (ASPA), France. The station is located in the outskirts of the city in an area characterised by large low-rise buildings (LCZ 8). Few or no trees are present and the land cover is mostly paved or hard-packed. Air temperature is measured with a platinum resistance thermometer (HMP35D, Vaisala) with radiation shield.
The site of Basel Binningen (BBIN) is operated by the Swiss Federal Office of Meteorology and Climatology (MeteoSwiss). The station is located in a rural/agricultural patch (LCZ D) at the south-western outskirts of the city. Low-rise residential areas surround the patch to the west, north and east. The surface underneath the sensor is natural (grass) and scattered low-rise buildings are present at 15-20 m distance. The site measures temperature with a thermometer with radiation shield and active ventilation (Rotronic HC2A-S3, Meteolabor). The site of Basel Lange Erlen (BLER) is operated by the MRC-Lab as well.
The station is located in a rural/agricultural patch (LCZ D) at 6 km distance from the city core. The surface underneath the sensor is natural (grass). Given the distance from the city centre and the vegetated environment, the site of BLER is used as rural reference in this study. The station used the same instruments as the site BKLI for air temperature, relative humidity and radiation fluxes.

### 3.4  Satellite observations

Thermal infrared (TIR) data from the Landsat 7 are used to evaluate the modelled land surface temperatures ($LST_{mod}$). Landsat acquires TIR data with a spatial resolution of 60 m. In contrast to other TIR satellite data (e.g. MODIS or SEVIRI), this resolution is fine enough to support high-resolution applications. Due to its small field of view, Landsat TIR retrievals are not affected by thermal angular anisotropy (Hu et al., 2016). During the investigation period, the Landsat 7 scene from 5 July 2015 was selected for the evaluation. No other cloud-free scenes from Landsat 7 or 8 were available during this period. Specifically,
the Landsat 8 overpass of 4 July 2015 had to be discarded due to the presence of cirrus clouds.

The satellite surface temperature ($LST_{sat}$) was derived from satellite data by correcting and converting the TIR signal received by the satellite sensor. The $LST_{sat}$ calculation follows the approach described in Mussetti et al. (2019). We used the radiative transfer model MODTRAN (MODerate resolution atmospheric TRANsmission) to estimate the atmospheric transmission, the long-wave upwelling or atmospheric path radiance and the long-wave downwelling or sky radiance (Abreu





and Anderson; Tardy et al., 2016). Atmospheric profiles from COSMO-BEP-Tree (STD simulation) were used in MODTRAN replacing its standard profiles. MODTRAN was used to transform the top-of-atmosphere radiance to surface-leaving radiance, which was further converted to LST by inverting the Planck function (Coll et al., 2010). A modified vegetation-threshold approach by Sobrino et al. (2008) was used to estimate the surface emissivity. As the original approach does not account for

built-up areas, a lookup table for dark and bright urban surfaces with a range of 0.95 to 0.97, based on emissivity values for typical urban fabrics (Baldridge et al., 2009), was applied (Mitraka et al., 2012). Using the above-described methods, the uncertainty in surface emissivity and atmospheric effects results in a total error of approximately 0.4 to 1.1 K (Jiménez-Muñoz and Sobrino, 2006).

The $\mathrm{LST_{sat}}$ has been re-mapped to the model grid of COSMO-BEP-Tree using an average re-sampling method, i.e. the

average of all contributing pixels was computed (McInerney and Kempeneers, 2015). This re-sampling method ensures the conservation of the mean values.

$\mathrm{LST_{mod}}$ was calculated combining the temperatures of roofs, streets, natural surfaces according to their respective fractions as:

$$\mathrm{LST_{mod}} = \sum_{d=1}^{N_d} T_{\mathrm{str}} f_{\mathrm{dir}} f_{str} (1 - CC) + \sum_{d=1}^{N_d} \overline{T_{\mathrm{veg}}} f_{\mathrm{dir}} f_{str} CC + \sum_{d=1}^{N_d} (\sum_{h=1}^{N_h} T_{\mathrm{roof}} f_{roof}) f_{\mathrm{dir}} f_{bld} + T_{\mathrm{nat}} \mathrm{f_{nat}}, \tag{14}$$

where $T_{\mathrm{str}}$ is the temperature of the streets (function of canyon direction ($_d$)), $CC$ is the effective canopy cover from street trees, $\overline{T_{\mathrm{veg}}}$ is the vertically averaged temperature of the leaves of the street trees, $T_{\mathrm{roof}}$ is the temperature of the roofs (function of ($_d$) and height ($_h$)) and $T_{\mathrm{nat}}$ is the temperature of the natural tile. $CC$ is estimated from the leaf area density $L$ using the relationship provided by Klingberg et al. (2017).

### 3.5 Performance metrics

The model performance was evaluated against the observations using the following metrics: root-mean-square error (RMSE), systematic root-mean-square error ($\mathrm{RMSE}_s$), unsystematic root-mean-square error ($\mathrm{RMSE}_u$), mean bias error (MBE) and coefficient of determination ($r^2$). This choice of metrics follows the recommendations of Willmott et al. (1985) and ensures comparability with previous urban climate model evaluations (e.g., Schubert and Grossman-Clarke, 2014; Oleson et al., 2008; Grimmond et al., 2011). The definition of the indices is available in Willmott et al. (1985); Willmott (1981).

## 4 Results

### 4.1 Evaluation against measurements from an urban flux tower

The comparison between model simulations and observations above the urban canopy layer at the BKLI flux tower site is shown in Fig. 5. Only the results corresponding to the reference simulation (STD) and to the simulation without street trees (LA0) are shown (see Table 1 for further details on the simulation set-ups). The statistical scores are given in Table 4. If not

specified, we will refer to the STD simulation in the text below.



**Figure 5.** Comparison between observations (Obs) and model simulations (STD, LA0) of air temperature (a), specific humidity (b), wind speed (c), sensible heat flux (d) and latent heat flux (e) at the site BKLI (38 m tall flux tower) during the selected simulation period (discarding the first 5 days as spin-up). Black dots indicate the observations. Blue and red lines indicate the results from the STD and LA0 model runs, respectively. The shaded areas represent the range of variability within the period for the observations and STD simulation only. Specifications of the model set-ups can be found in Section 3.1.

The model simulates the evolution of air temperature very well during the evaluation period (Figure 5a) although a slight overestimation with a MBE of 0.51 K and a systematic RMSE of 0.51 K was found. The overestimation occurs mostly during night-time, an issue that was observed also in other urban climate studies with COSMO (Schubert and Grossman-Clarke, 2014; Wouters et al., 2016) and other models (e.g. Lee et al., 2011). The night-time overestimation of air temperature is attributed to
5   the inability of COSMO to represent very stable boundary layer conditions (Cerenzia, 2017). The impact of street trees on the air temperature above the canopy layer is found to be negligible, giving the very small differences between STD and LA0.



**Table 4.** Root-mean-square errors (RMSE), mean-bias errors (MBE) and coefficient of determination ($r^2$) of the simulated air temperatures ($T_{\mathrm{air}}$), specific humidity ($q$), wind speed ($u$), sensible heat flux ($Q_{\mathrm{H}}$) and latent heat flux ($Q_{\mathrm{E}}$) at the BKLI site (38 m above ground) during the period 22 June 2015 - 9 July 2015 (discarding the first 5 days as spin-up). In addition to the total RMSEs (T), the systematic (S) and the unsystematic (U) RMSEs are listed. The statistics are provided for the standard simulation (STD) and the simulation without street trees (see Table 1 for details on the simulation set-up).

|  |  | RMSE (T/S/U) | MBE | $r^2$ |
|---|---|---|---|---|
| $T_{\mathrm{air}}$ (K) | STD | 1.04 / 0.51 / 0.9 | 0.51 | 0.97 |
|  | LA0 | 1.1 / 0.58 / 0.93 | 0.58 | 0.97 |
| $q$ (g kg$^{-1}$) | STD | 1.52 / 1.13 / 1.02 | -1.09 | 0.74 |
|  | LA0 | 1.61 / 1.24 / 1.03 | -1.2 | 0.73 |
| $u$ (m s$^{-1}$) | STD | 1.31 / 0.8 / 1.03 | 0.61 | 0.7 |
|  | LA0 | 1.62 / 1.1 / 1.19 | 0.87 | 0.69 |
| $Q_{\mathrm{H}}$ (W m$^{-2}$) | STD | 40.1 / 7.72 / 39.3 | 7.7 | 0.88 |
|  | LA0 | 74.8 / 50.6 / 55.2 | 47.0 | 0.84 |
| $Q_{\mathrm{E}}$ (W m$^{-2}$) | STD | 25.1 / 6.86 / 24.1 | 5.31 | 0.70 |
|  | LA0 | 37.8 / 36.9 / 8.57 | -32.2 | 0.74 |

The evolution of specific humidity $q$ throughout the simulation is generally well reproduced (Figure 5a-b). The mean value of $q$ is underestimated (MBE = -1.09 g kg$^{-1}$), which explains a large fraction of the systematic RMSE (RMSE$_s$ = 1.13 g kg$^{-1}$). A similar pattern is found at the rural site of BLER (available in the Supplement), indicating that the bias is not related to the urban scheme. The bias would be consistent with a general underestimation of soil moisture in COSMO as reported by

Davin et al. (2011). Another possible source of the bias may be related to the boundary conditions, with insufficient humidity advected from the boundaries of the domain. Street trees produce a small increase in relative humidity bringing the model in slightly closer agreement with the observations. This is quantified by a reduction in the systematic component of RMSE and MBE compared to LA0 of about 0.1 g kg$^{-1}$ (about 6 and 8 % of the entire RMSE and MBE, respectively).

The model simulates well the evolution of wind speed ($u$) during the evaluation period (Figure 5c-d), but overestimates $u$

during a few days and especially in the afternoon. The origin of the overestimation is unclear. It may be related to either a misrepresentation of the synoptic wind from the driving boundary conditions and an overestimation of the thermally driven flow induced by the surface temperature gradient between the city and the surroundings. Nevertheless, the comparison with observations shows a small bias (MBE = 0.61 m s$^{-1}$), lower than previous urban studies with COSMO (e.g. Schubert and Grossman-Clarke, 2014), and the unsystematic error dominates. Street trees reduce the simulated $u$ and slightly improve the

agreement with the observations. RMSE and MBE are reduced compared to LA0 of 0.31 and 0.26 m s$^{-1}$, respectively.

The sensible heat flux ($Q_{\mathrm{H}}$) follows very well the observations during the evaluation period (Figure 5e-f) showing almost no bias (MBE = 7.7 W m$^{-2}$) but a RMSE of 40.1 W m$^{-2}$ with a large unsystematic component. These errors are comparable to, or smaller than those obtained in a recent inter-comparison of UCMs (Grimmond et al., 2011). The representation of street



trees substantially improves the agreement with the observations. Street trees reduce simulated $Q_\mathrm{H}$ by as much as 100 W m$^{-2}$ during mid-day. The improvements in RMSE and MBE compared to LA0 were as large as 34.7 and 39.3 W m$^{-2}$, respectively. Additionally, an improvement of 0.06 in $r^2$ was also found.

The evolution of latent heat flux ($Q_\mathrm{E}$) is generally well reproduced by the model (Figure 5g-h), but the agreement is better

during the initial and final phase of the heatwave when the synoptic forcing was stronger and wind speeds correspondingly higher. During the central part of the evaluation period, a slight overestimation is found. Overall, the model showed a very small MBE of about 5 W m$^{-2}$ and a RMSE of 25.1 W m$^{-2}$. Again, the scores are comparable to, or better than those obtained in a recent inter-comparison of UCMs (Grimmond et al., 2011). The slight overestimation during the central part of the simulation period could be caused by a limitation in soil water availability that may have been present in reality but was not considered

in the simulations. Street trees had a very large impact on $Q_\mathrm{E}$ and improved the agreement with the observations substantially. RMSE and MBE improved compared to LA0 by 12.7 and 26.9 W m$^{-2}$, respectively.

### 4.2   Evaluation against surface stations

Within the urban canopy layer, the model was evaluated in terms of the 2-m air temperature ($T_\mathrm{2m}$) and canopy layer urban heat island (UHI) intensity. The UHI intensity is defined as $\mathrm{UHI}_i = T_i - T_\mathrm{rur}$ where $T_i$ is the 2-m air temperature at the urban site $i$

and $T_\mathrm{rur}$ is the 2-m temperature at the rural reference site BLER. The comparison between model simulations and observations in term of 2-m air temperature $T_\mathrm{2m}$ is shown in Fig. 6. Only the results corresponding to the reference simulation (STD) and to the simulation without street trees (LA0) are shown. If not specified, we will refer to the STD simulation in the text below.

The model simulation compares generally well with the observed $T_\mathrm{2m}$, especially in the urban area. The model errors are in the range of other urban climate studies (e.g. Schubert and Grossman-Clarke, 2014; Wouters et al., 2016). The difference

between model and observations is more pronounced at the site BFEL where the systematic part of the RMSE is substantially higher (RMSE$_s$ = 0.93 K) than at other sites. This difference may be related to exposure of the sensor to local influences. Notable is also the overestimation of temperatures at the rural sites BLER and BBIN (MBE = 1.36 K and 0.88 K, respectively), which is most pronounced at night-time. This is a well known issue of COSMO that occurs during very stable atmospheric conditions (Buzzi et al., 2011; Cerenzia, 2017) and has already been reported in previous studies (e.g. Schubert and Grossman-

Clarke, 2014).

The representation of street trees produces a slight decrease in daytime air temperature, particularly around noon (Figure 6a-b). This effect is more evident at sites with higher urban fraction ($f_\mathrm{urb}$) and leaf are index from street trees ($L_\mathrm{can}$). The representation of street trees slightly reduced the RMSE and MBE at all the sites (but BFEL) compared to LA0. When averaged over all sites, the reductions in RMSE and MBE were 0.03 and 0.11 K, respectively.

The comparison between model simulation and observations in terms of UHI intensity for the simulations STD and LA0 is shown in Fig. 7. The statistical scores are presented in Table 6 for the simulation STD only.

The model is able to capture the general features of the UHI, such as the daily evolution and the different magnitude of the UHI at the different sites, whereas the temporal variability throughout the period is not always represented. Generally, the UHI intensity is underestimated, especially during night-time. This is related to the overestimation of night-time air temperatures at

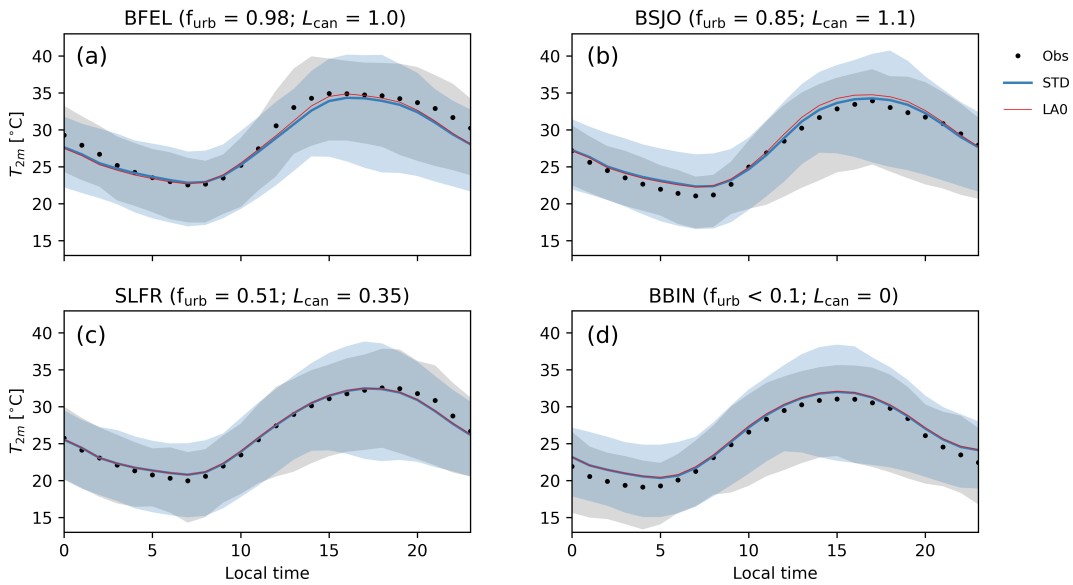

**Figure 6.** Comparison between period averaged daily profiles of observed (Obs) and model-simulated 2-m air temperature at the sites BFEL, BSJO, SLFR and BBIN during the selected period (discarding the first 5 days as spin-up). Black dots indicate the observations. Blue lines and red lines indicate the results from the STD and LA0 model runs, respectively. The range of variability within the selected period is shown as shaded area for Obs (grey) and STD (light blue).

the rural site BLER (MBE = 1.36 K for STD) and, to a lower extent, to the underestimation at the most urban sites (Table 5 and Supplement). The performance with respect to UHI intensity is better or comparable to similar studies (Wouters et al., 2016; Schubert and Grossman-Clarke, 2014).

The representation of street trees affects the simulated UHI intensity by producing a slight reduction in UHI intensity from late morning to early evening. The impact of street trees is more evident at sites with higher urban fraction and density of street trees. The representation of street trees did not improve the model performance in terms of UHI representation, given that the warm bias at the rural site dominates.

### 4.3   Evaluation against satellite observations of land surface temperature

Figure 8 compares the spatial distribution of simulated and remotely-sensed land surface temperature (LST) on 5 July 2015 at approx. 11:10 LT (10:10 UTC). Statistical scores are shown in Table 7.

Overall, a good agreement between simulations and observations was obtained in terms of the domain averaged model bias and errors. As shown in Fig. 8, the spatial pattern of LST is generally well captured by the model indicating that the urban and rural land use variations are well represented. However, the model is found to underestimate the maximum LSTs and to overestimate the minimum LSTs at the outskirt of the urban area (Figure 8d-e). Quantitatively, the model was found to have a small negative bias over the urban area (MBE = -0.39 K for STD) and a small positive bias when considering the entire domain



**Table 5.** Root-mean-square errors (RMSE), mean-bias errors (MBE) and coefficient of determination ($r^2$) of the simulated 2-m air temperature $T_{2m}$ at the sites BFEL, BSJO, SLFR, BBIN and BLER during the period 22 June 2015 - 9 July 2015 (discarding the first 5 days as spin-up). In addition to the total RMSEs (T), the systematic (S) and the unsystematic (U) RMSEs are listed. The statistics are provided for the standard simulation (STD) and the simulation without street trees (LA0).

| | 2-m air temperature (K) | | |
|---|---|---|---|
| | | RMSE (T/S/U) | MBE | $r^2$ |
| BFEL | STD | 1.66 / 0.93 / 1.37 | -0.85 | 0.94 |
| | LA0 | 1.59 / 0.72 / 1.41 | -0.7 | 0.94 |
| BSJO | STD | 1.39 / 0.56 / 1.27 | 0.56 | 0.95 |
| | LA0 | 1.55 / 0.81 / 1.32 | 0.77 | 0.95 |
| SLFR | STD | 1.03 / 0.19 / 1.01 | 0.04 | 0.97 |
| | LA0 | 1.03 / 0.18 / 1.02 | 0.08 | 0.97 |
| BBIN | STD | 1.39 / 0.90 / 1.07 | 0.88 | 0.96 |
| | LA0 | 1.43 / 0.99 / 1.04 | 0.98 | 0.97 |
| BLER | STD | 2.17 / 1.41 / 1.65 | 1.36 | 0.91 |
| | LA0 | 2.2 / 1.45 / 1.65 | 1.4 | 0.91 |
| All[a] | STD | 1.53 / 0.80 / 1.27 | 0.40 | 0.95 |
| | LA0 | 1.56 / 0.83 / 1.29 | 0.51 | 0.95 |

[a] average over all sites.

(MBE = 0.34 K for STD). The RMSE was found to be lower in the urban area (RMBS = 2.71 K for STD) than in the entire domain. However, a relatively large component of RMSE was found to be systematic. The coefficient of determination was found to be only quite low ($r^2$ = 0.58 for STD). Only minor differences between STD and LA0, with STD being slightly colder over the urban area.

The moderate $r^2$ and the relatively high component of systematic RMSE over urban areas may be explained by the fact that the same material properties (albedo, emissivity, heat capacity and heat conductivity) were used for the whole city. Therefore, spatial variations in material properties, e.g. between industrial and residential buildings, were not represented. This likely limits the model's ability to represent intra-urban gradients of surface temperature and, consequently, the related components such as sensible heat fluxes.

The explicit representation of street trees did not improve the agreement with satellite observations of land surface temperature. However, this observation should be taken with care given the uncertainty in the satellite observations of the order of 1 K.

    Studies comparing simulated and remotely sensed surface temperatures over cities are rather scarce. We were only able to quantitatively compare our results with the studies of Wouters et al. (2016) and Hu et al. (2014). Other studies only performed a

qualitative comparison (Miao et al., 2009; Giannaros et al., 2013; Li and Bou-Zeid, 2013; Ramamurthy et al., 2017). The MBE



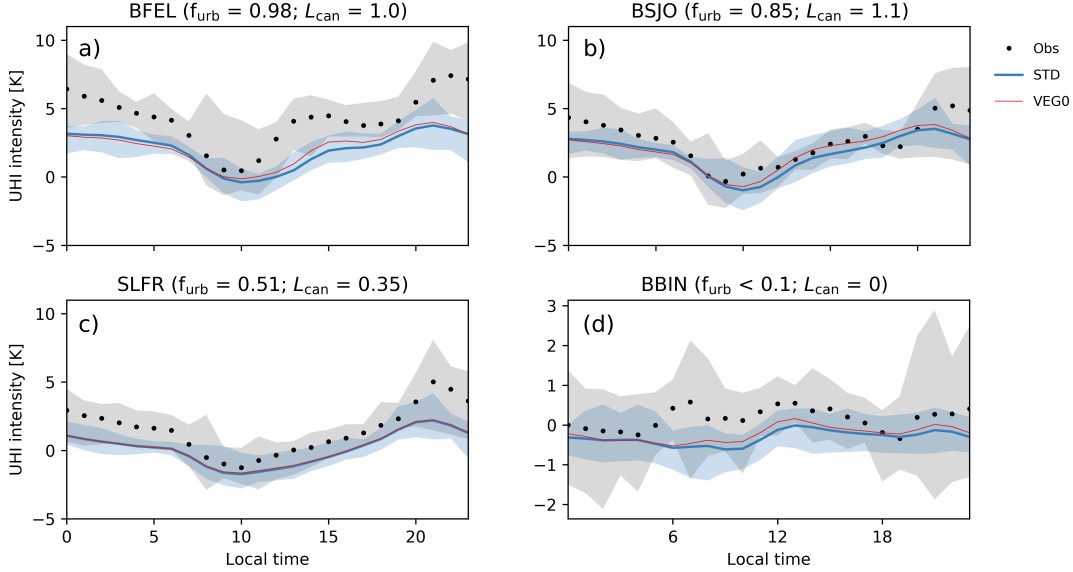

**Figure 7.** Comparison between period-averaged daily profiles of observed (Obs) and model-simulated 2-m urban heat island (UHI) intensity at the sites BFEL, BSJO, SLFR and BBIN during the evaluation period. The site BLER is used as rural reference. Black dots indicate the observations. Blue and red lines indicate the results from the STD and LA0 model runs, respectively. The range of variability within the selected period is shown as shaded area (only for Obs and STD).

found in the present study is lower than what was found by Wouters et al. (2016) and Hu et al. (2014) for most of the conditions (different urban fractions and rural). Like the present study, Hu et al. (2014) found a general overestimation of daytime ST in their model while Wouters et al. (2016) reported an opposite behaviour. Wouters et al. (2016) and Hu et al. (2014) did not use other quantitative statistical metrics. Therefore, a comparison for RMSE and $r^2$ was not possible.

## 4.4 Model sensitivity to tree parameters

Figure 9 shows the model sensitivity of 2-m air temperature (called here $T_{\mathrm{air}}$), street surface temperature $T_{\mathrm{str}}$, specific humidity $q$ and wind speed $u$ to varying values of tree parameters (LA+, LA-, SC+, SC-). The sensitivity $S_{\phi,i}$ is defined here in relative terms normalised by the amplitude of the difference between a simulation with and without street trees:

$$S_{\phi,i} = \frac{\overline{\phi_i} - \overline{\phi_{\mathrm{STD}}}}{\left|\overline{\phi_{\mathrm{LA0}}} - \overline{\phi_{\mathrm{STD}}}\right|}, \tag{15}$$

where $\phi$ is a generic scalar field ($T_{\mathrm{air}}$, $T_{\mathrm{str}}$, $q$, $u$), $\overline{\phi}$ represents the spatial and temporal average of the scalar field, and $i$ represents the experiment as in Table 1. Generally, the model was more sensitive to changes in leaf area density (LA+, LA-) than to changes in stomatal conductance (SC+, SC-). During daytime (10-18 local time), $T_{\mathrm{air}}$ and $T_{\mathrm{str}}$ were the variables that were most sensitive to the choices of street tree parameters, whereas $u$ was the least sensitive. At night (21-5 local time), conversely, the variable with the largest sensitivity was $q$.





**Table 6.** Root-mean-square errors (RMSE), mean-bias errors (MBE) and coefficient of determination ($r^2$) of the simulated 2-m urban heat island (UHI) intensity (K) at the sites of BFEL, BSJO, SLFR and BBIN during the period 22 June 2015 - 9 July 2015 (discarding the first 5 days as spin-up). The site of BLER is used as rural reference. In addition to the total RMSEs (T), the systematic (S) and the unsystematic (U) RMSEs are listed.

| 2-m UHI intensity (K) | | | | |
|---|---|---|---|---|
| | | RMSE (T/S/U) | MBE | $r^2$ |
| BFEL | STD | 2.7 / 2.53 / 0.95 | -2.26 | 0.57 |
| | LA0 | 2.6 / 2.43 / 0.92 | -2.12 | 0.56 |
| BSJO | STD | 1.56 / 1.2 / 1.0 | -0.86 | 0.54 |
| | LA0 | 1.53 / 1.12 / 1.04 | -0.65 | 0.48 |
| SLFR | STD | 1.8 / 1.63 / 0.75 | -1.37 | 0.67 |
| | LA0 | 1.77 / 1.6 / 0.75 | -1.33 | 0.67 |
| BBIN | STD | 0.93 / 0.86 / 0.36 | -0.48 | 0.43 |
| | LA0 | 0.86 / 0.78 / 0.37 | -0.38 | 0.45 |
| All[1] | STD | 1.66 / 1.42 / 0.82 | -1.18 | 0.55 |
| | LA0 | 1.60 / 1.38 / 0.81 | -1.10 | 0.54 |

[1] average over all sites.

**Table 7.** Root-mean-square errors (RMSE), mean-bias errors (MBE) and coefficient of determination ($r^2$) of the simulated (STD) land surface temperature (LST) [K] against the Landsat 7 satellite observations on the 5 July 2015 at approx. 11:10 LT (10:10 UTC). In addition to the total RMSEs (T), the systematic (S) and the unsystematic (U) RMSEs are listed. "Urb" represents the comparison only at the urban grid cells ($f_{urb} > 0.1$). Grid cells contaminated by water or clouds have been removed.

| Surface Temperature | | | | |
|---|---|---|---|---|
| | | RMSE (T/S/U) | MBE | $r^2$ |
| All (N=5853) | STD | 3.14 / 2.36 / 2.07 | 0.34 | 0.6 |
| | LA0 | 3.08 / 2.25 / 2.1 | 0.53 | 0.63 |
| Urb (N=1930) | STD | 2.71 / 2.36 / 1.34 | -0.39 | 0.58 |
| | LA0 | 2.61 / 2.25 / 1.31 | -0.03 | 0.62 |

An increase in the leaf area density from street trees (LA+) produced a reduction in daytime $T_{air}$, $T_{str}$ and $u$, and an increase in $q$. The reduction in $T_{air}$ and $T_{str}$ can be explained by the increase in interception of short-wave radiation from the denser foliage layer (see Eq. (1)) and, consequently, by the reduced amount of short-wave radiation reaching the wall and street surfaces within the canyon. The reduction in $u$ can be explained by the augmented drag force excerpted by the denser foliage layer (see Eq. (2)). The increase of $q$ can be explained by the larger amount of leaf area available for transpiration (see Eq. (5)).

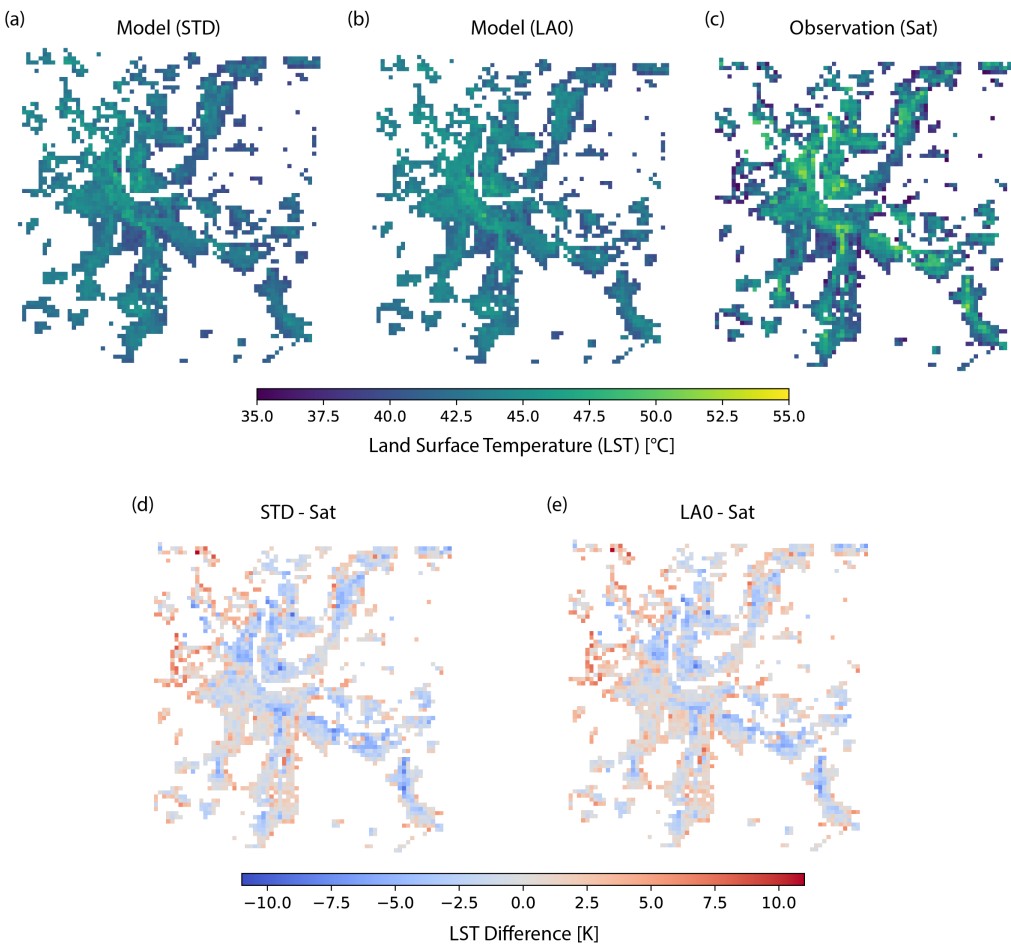

**Figure 8.** Spatial distribution of land surface temperature (LST) [$^\circ$C] on 5 July 2015 at approx. 11:10 LT (10:10 UTC). (a,b) show the simulated LST with STD and LA0 configurations, respectively; (c) shows the observed LST from Landsat 7 satellite; (d,e) show the LST differences between simulated and observed LST for the STD and LA0 configurations, respectively. Grid cells contaminated by the presence of the river or clouds have been removed.

During night-time, the sensitivity experiment LA+ produced a marked increase in $q$ and $T_{\mathrm{str}}$. $T_{\mathrm{air}}$ remained approximately constant while $u$ decreased. The responses of $q$ and $u$ follow the same mechanisms of the daytime response. The response of $T_{\mathrm{str}}$, in contrast, is opposite to its daytime response. At night, enhanced interception of long-wave radiation emitted from the internal surfaces of the canyon (walls and street) with increased leaf area density reduces the cooling of the street surface.

5 Although enhanced tree foliage cools the street surfaces during the day due to shading, the heat capacity and corresponding thermal inertia of the street surfaces seems to be not sufficient to carry over this effect into the night-time to compensate for the reduced long-wave radiative cooling. Such an increase in night-time surface temperatures due to street trees, although

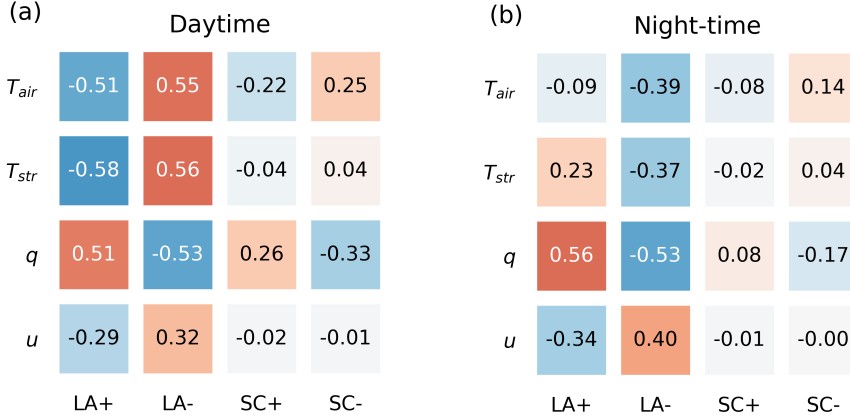

**Figure 9.** Model sensitivity $S$ of 2-m air temperature $T_{air\,2m}$, street surface temperature $T_{str}$, specific humidity $q$ and wind speed $u$ to varying values of tree parameters (LA+, LA-, SC+, SC-). Values of $S$ represent a spatial (over all urban grid cells) and temporal (over whole simulation period) average. Daytime corresponds to 10-18 local time, night-time to 21-5 local time.

somewhat counter-intuitive, has in fact been reported previously in observation-based studies (e.g. Shashua-Bar et al., 2009; Coutts et al., 2016).

A reduction in the leaf area density (LA-) generally produced an opposite response to that of LA+. An exception is the reduction of $T_{air}$ during night-time.

An increase in the stomatal conductance (SC+) produced an increase in daytime $q$ and a decrease in $T_{air}$. A decrease (SC-) produced an opposite response. These effects are due to an increase in the source of moisture (see Eq. 5) and latent heat flux $Q_E$ (see Eq. 4) as a response to the larger value of stomatal conductance $g_s$. The increase in $Q_E$ causes a reduction in $Q_H$ that consequently impacts $T_{air}$. $T_{str}$ and $u$ were found to be insensitive to changes in the stomatal conductance. This is expected since, in contrast to leaf area density, stomatal conductance has no influence on short-wave radiation or drag. During night-time, the model was found to be rather insensitive to changes in stomatal conductance. This is not surprising giving the low value of night-time stomatal conductance in the standard run (STD).

## 5 Future work

We acknowledge that there are a number of potential limitations in the current model system that need to be addressed in future studies.

BEP-Tree, and therefore COSMO-BEP-Tree, employ a rather simplistic *stomata model* for representing the transpiration from street trees. In particular, constant and spatially homogeneous values of daytime and night-time stomatal conductance were employed ignoring the potentially important impact of water scarcity and the dependence on environmental conditions (e.g. air temperature and solar irradiation). In order to improve the representation of the transpiration from street trees and





therefore the modelling of the associated latent heat fluxes, a mechanistic stomata model (e.g. Damour et al. (2010)) will have to be implemented.

In this study, a homogeneous and constant value of *clumping index* ($\Omega$) was used. The value was chosen from a measurement over a natural environment that resembles the urban environment in terms of openness and heterogeneity of the tree canopy.

In order to improve urban climate simulations with street trees, a methodology to estimate the spatial distribution of $\Omega$ over urban environments needs to be established. A future implementation of the clumping effects should also consider the temporal changes of $\Omega$ that is known to vary with the solar zenith angle (Ryu et al., 2016).

The *coupling approach* employed in this work to link COSMO-CLM and BEP-Tree is based on model integration (also called joint coupling by Brandmeyer and Karimi (2000)) rather than on an independent coupler. The model integration approach has

the advantage of maximising the computational performance by reducing the information exchanged. As a drawback, it does not support independent model updates. In other words, every update in the standalone version of each individual model (COSMO-CLM and BEP-Tree) potentially requires additional effort in re-building the coupled version. In order to allow flexible coupling between future versions of COSMO-CLM and BEP-Tree, the use of a coupler would be interesting. As an example, COSMO-CLM has been already used with the OASIS coupler (Craig et al., 2017) to couple with different land surface models (Will

et al., 2017).

Even if generally comparable with other models, the *performance* of COSMO-BEP-Tree still has room for improvements. The model evaluation against urban flux tower measurements revealed a systematic underestimation of specific humidity ($q$). The source of this bias needs to be better investigated by, for instance, (a) analysing the soil moisture content and (b) evaluating $q$ at the model boundaries provided by another model. The model evaluation at 2-m reveals an overestimation of

the air temperature at the rural sites especially at night. This issue has been already found in other studies (e.g. Schubert and Grossman-Clarke, 2014) and is attributed to the inability of the model to reproduce very stable atmospheric conditions (Buzzi et al., 2011). More recent versions of the COSMO model (from version 5.4a onward) include a revised turbulence scheme that has the potential to better represent such conditions.

In the current study, spatially homogeneous values of *material properties* of roof, wall and street were used. Therefore, the

impact of the spatial variation of material properties between areas with different building types (e.g. old town vs. industrial areas) could not be reproduced. Further versions of UCPgenerator need to consider different urban classes with distinct material properties. This could be achieved, for example, by making use of the data from the World Urban Database and Access Portal Tools (WUDAP, Ching et al. (2018)).

## 6 Conclusions

An urban climate model with explicit representation of street trees was developed featuring a two-way coupling between the vegetated urban canopy model BEP-Tree (Krayenhoff et al., 2019) and the mesoscale weather and climate model COSMO-CLM (COSMO). The coupled model, named COSMO-BEP-Tree, mechanistically represents the heat, momentum, turbulence and moisture interactions between street trees, urban elements and the atmosphere during realistic weather and climate simu-





lations. COSMO-BEP-Tree comes with a tool to pre-process all data required by BEP-Tree in addition to the standard inputs for COSMO.

The model's performance and applicability were demonstrated for the urban area of Basel during a three week heatwave event in June-July 2015. The model evaluation made use of measurements from an urban flux tower, a network of surface

stations, and land surface temperature observations from the Landsat 7 satellite. The application demonstrated that the model successfully captures essential features of the urban heat island during such a heat wave event and that city-wide interactions between street trees, urban elements and the atmosphere are represented in a realistic way.

In particular, the model captured the day-to-day as well as the mean diurnal evolution of air temperature, relative humidity, wind speed, sensible and latent heat fluxes relatively well at all observation sites. The comparison with measurements above

the urban canopy layer revealed very small model biases and errors for air temperature, wind speed, sensible and latent heat fluxes. The model underestimated the specific humidity and slightly overestimated wind speed in the afternoon.

The comparison with measurements of air temperature within the urban canopy layer ($T_{2m}$) revealed a generally good model performance. At rural sites, the model showed an overestimation of $T_{2m}$ as already reported in previous urban climate studies using the COSMO model. The model was able to reproduce the diurnal cycle of the canopy layer urban heat island effect (UHI)

generally well while its magnitude was underestimated during night-time. This issue was largely due to the warm night-time bias at the rural reference site. At the most urban site, it is additionally explained by an underestimation of the night-time temperature.

In terms of the spatial pattern of the land surface temperature (LST), the model was found to have a relatively small RMSE and very small MBE compared to the satellite observations. However, the spatial variability over the urban area and especially

some local peaks in LST were not fully captured by the model, which could be related to the use of spatially homogeneous material properties.

The inclusion of street trees in the model generally improved the performance against observations of $Q_H$, $u$, $q$ and $Q_E$. In the case of $T_{2m}$ this was found only in some of the evaluation sites. The representation of street trees had a large effect on sensible and latent heat fluxes and wind speed but only a minor influence on specific humidity and air temperature above the

urban canopy layer. Street trees were found to reduce $Q_H$ and $u$, and to increase $q$ and $Q_E$. Within the canopy layer, a slight decrease in $T_{2m}$ during daytime was found when including street trees, and a very small increase in $T_{2m}$ during night-time. This behaviour is in agreement with measurement studies over street canyons with trees (Shashua-Bar et al., 2009; Coutts et al., 2016). In terms of UHI intensity, this results in a slight reduction during daytime and a very small increase during night-time.

The model responded realistically to changes in the street trees parameters: leaf area density and stomatal conductance.

During daytime, a 50% increase in leaf area density produced a reduction in air temperature, street temperature and wind, and an increase in humidity. At night, the same 50% increase in street trees density produced mainly an increase in humidity. A 50% increase in stomatal conductance produced an increase in humidity and a decrease air temperature.

This study demonstrates the ability of COSMO-BEP-Tree to simulate the impact of street trees for whole cities in a realistic way, by considering the multiple interactions between trees, urban elements and the atmosphere from the scale of individual

street canyons to the scale of regional weather patterns. By including the effect of street trees, COSMO-BEP-Tree significantly



enhances the modelling capabilities with respect to its predecessor COSMO-DCEP, adding an important dimension to the study of UHI mitigation and climate change adaptation strategies. Furthermore, the improved agreement with observations clearly underlines the importance of considering street trees in such simulations.

*Code and data availability.* COSMO-CLM is freely available for scientific usage to the members of the CLM-Community. The procedure
5  to become a member is available on the community website (https://www.clm-community.eu/) or by contacting the coordination office (clm-coordination@dwd.de). The code of COSMO-BEP-Tree is made freely available to any user with a valid CLM-Community membership (contact Dominik Brunner (dominik.brunner@empa.ch). All research data and scripts to produce figures and analysis are archived on Zenodo (Mussetti, 2019a). UCPgenerator v1.0 can be downloaded from the GitHub repository (Mussetti, 2019b). The LUCY model, used to calculate the anthropogenic heat flux, can be downloaded from Grimmond et al. (2018). BEP-Tree as standalone model can be obtained by contacting
10 Scott Krayenhoff (skrayenh@gmail.com). MODTRAN is a commercial software: http://modtran.spectral.com/ (accessed 26-08-2019).





**Table A1.** Urban canopy parameters required as input to COSMO-BEP-Tree.

| Variable | Description | Dimension | NetCDF Name |
|----------|-------------|-----------|-------------|
| $f_{urb}$ | Planar fraction of urban surfaces | lon, lat | FR_URB |
| $W_{can}$ | Average canyon width [m] | lon, lat, dir | W_STREET |
| $W_{bld}$ | Average building width [m] | lon, lat, dir | W_BUILD |
| $f_{dir}$ | Canyon direction distribution | lon, lat, dir | FR_UDIR |
| $\gamma$ | Probability to have a building | lon, lat, dir, hgt | FR_ROOF |
| $L_D$ | Leaf area density in the canyon column | lon, lat, dir, hgt | LAD_C |
| $L_{D,bld}$ | Leaf area density in the building column | lon, lat, dir, hgt | LAD_B |
| $\Omega$ | Clumping index | lon, lat | OMEGA_R, OMEGA_D[a] |

[a] BEP-Tree gives the option to define a separate clumping coefficient for the drag and turbulence parameterisations.

## Appendix A: UCPgenerator v1.0

UCPgenerator v1.0 is a python program that generates the additional input fields required by COSMO-BEP-Tree (Mussetti, 2019b). The list of the additional input field required by COSMO-BEP-Tree is provided in Table A1. First of all, the urban fraction $f_{urb}$, defined as the planar fraction of urban surfaces in a model grid cell, is estimated from the given imperviousness

dataset (see Section 2.4 for further details). A threshold method is used estimate the average urban fraction in the COSMO-BEP-Tree grid cell (function `urbfrac`). `urbfrac` performs also the reference system transformation.

  The parameters that characterise the canyon geometry are then estimated in the function `cangeom` from the building dataset. Planar and frontal area density are calculated according to Grimmond and Oke (1999) as:

$$\lambda_p = \frac{\overline{A_P}}{\overline{A_T}} \tag{A1}$$

and

$$\lambda_f = \frac{\overline{A_F}}{\overline{A_T}}, \tag{A2}$$

where $\lambda_p$ is the planar area density, $\lambda_f$ is the frontal area density, $\overline{A_P}$ is the plan area of building elements, $\overline{A_T}$ is the total horizontal surface area. Canyon and building widths are calculated according to Martilli (2009) as:

$$W_{bld} = \frac{\lambda_p}{\lambda_f} h \tag{A3}$$

and

$$W_{can} = (\frac{1}{\lambda_p} - 1)\frac{\lambda_p}{\lambda_f}, \tag{A4}$$

where $W_{can}$ is the canyon width [m], $W_{bld}$ is the building width [m] and $h$ is the average building height [m].



Each building in the building dataset is clustered according to heigh and direction classes that can be set by the user through the namelist of COSMO-BEP-Tree. The vertical distribution of buildings $\gamma$ and the distribution of canyon directions $f_{\mathrm{dir}}$ are determined from the above mentioned clustering.

The sub-routine `veget` finally computes the grid cell averaged values of $L_{\mathrm{D,can}}$ and $L_{\mathrm{D,nat}}$. It is designed to read as
input a LIDAR based dataset containing the height distribution of street trees. Making assumptions on the specific leaf area density inside an average tree ($L_{\mathrm{D,loc}}$), the routine calculates the total $L_{\mathrm{D}}$ in each model grid cell. If a dataset of all trees (e.g. including trees in parks) is available, the sub-routine identifies the street trees by using $f_{\mathrm{urb}}$. Specifically, trees located over mostly impervious surface ($f_{\mathrm{urb}} > 0.5$) are classified as street trees (contributing to $L_{\mathrm{D,can}}$). The rest of the trees are classified as natural trees and contribute to $L_{\mathrm{D,nat}}$. The routine also supports user defined modifications to the $L_{\mathrm{D}}$ to simulated greening
or clearing scenarios.

UCPgenerator v1.0 requires the 3D building dataset to be in the format of ESRI shapefiles (.shp) with a level of detail (LoD) of 1. In LoD 1 datasets, buildings are represented as prismatic blocks without information on the roof structure (Kolbe et al., 2005).

*Author contributions.* GM conceived the work, performed the simulations and analysed the results. SH and GM implemented the coupling
between COSMO and BEP-Tree. GM, DB, SH, ESK interpreted the results. ESK and SS provided the source codes of BEP-Tree and COSMO-DCEP, respectively. CF, RV and AW provided some of the input data and observations. DB, SH, JA and JC supervised the work. GM wrote the manuscript with contributions from all the co-authors.

*Competing interests.* The authors declare that they have no conflicts of interest.

*Acknowledgements.* The study was funded by the Swiss Federal Laboratories for Materials Science and Technology (Empa). The Center
for Climate Systems Modeling (C2SM) at ETH Zurich is acknowledged for providing technical and scientific support We thank the MRC-Lab of the University of Basel, the environmental agency of Basel, the environmental agency of Alsace and MeteoSwiss for providing meteorological observations. MeteoSwiss is also acknowledged for providing the meteorological analysis. Landsat-7 image is courtesy of the U.S. Geological Survey. The use of Empa's computational resources is greatly acknowledged. We thank Dominik Strebel for his work on the building dataset. We thank Arthur Gessler and Harald Bugmann for discussions about the values of stomatal conductance.





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
