# Peer review of "COSMO-BEP-Tree v1.0: a coupled urban climate model with explicit representation of street trees"

_Geoscientific Model Development, 2019_

## Referee Comment (RC1) · Anonymous Referee #1 · 16 Oct 2019

**GMD-2019-220 Review**

Realistic representation of urban vegetation is an important step to improving model capabilities and performance. In this paper, a vegetated urban canopy model incorporating street trees (BEP-Tree) is coupled with a mesoscale model (COSMO). The authors briefly describe the relevant parts of the models and their combination, before evaluating the combined model against a range of observations. The work is well-presented, with a clear structure and, mostly, an appropriate level of detail (although in some places more discussion would be beneficial). The inclusion of street trees within the model is shown to make generally small improvements to the meteorology but substantial improvements to the sensible and latent heat fluxes. The coupled model will be useful for applications concerning urban greening as well as for more general studies which will benefit from better representation of vegetation processes.

The paper is very well-presented overall. There are a few consistency issues which need to be addressed and a little more explanation would be beneficial in parts. However, I recommend this publication following minor revisions.

Main points

- P8 L15: The lack of interaction with soil moisture seems to be the major weakness in this study. More discussion would be beneficial. Why was the decision made not to implement this interaction? Can the authors comment on the effect of this decision? How realistic is it to assume transpiration in urban areas is never limited by soil moisture (I would think not very realistic)?

- P10 L6: It is not clear how $T_g$ relates to $T_{nat}$ or $T_s$ relates to $T_{urb}$

- P11 L5: Provide some justification for the statement about Basel's 'abundant green infrastructures' as the statistics and Fig 3b do not suggest a particularly vegetated city.

- P11 L16: Can the authors add whether any spurious structures occurred (as in other studies, e.g. Salamanca et al. 2012) given the small horizontal grid spacing used is in the 'grey zone'?
  Salamanca F, Martilli A, Yague C (2012) A numerical study of the urban heat island over Madrid during the DESIREX (2008) field campaign with WRF and an evaluation of simple mitigation strategies, IJC 32: 2372–2386 DOI: 10.1002/joc.3398

- P12 Fig 3: Station WEIL does not seem to be used anywhere in the manuscript – delete from the map

- P13 L3-4: Two sentences providing a general description of the imperviousness dataset and the buildings dataset would be helpful (i.e. are these fractions of impervious surface cover or degree of imperviousness; is it building height or building material or building use; at what resolution?)

- P15 Table 3: Latent heat flux seems to be denoted $Q_L$ here and $Q_E$ later in the manuscript – make consistent in all text, tables and figures. Fluxes also appear in the row for BLER – presumably this is incorrect?

- P15 L21-22: 'net' → 'four-component' as the CNR4 measures all four radiation components individually. Also, the radiation measurements are mentioned here but not used in the evaluation (e.g. in Fig 5). Why not? Can the authors comment on the results for the radiation components for STD and LA0 runs?

- P16 L4: Aren't there only five surface stations (depending on whether the WEIL data should be included or not)? Can the authors clarify here why no results from WEIL have been included?

- P17 L5: Here the emissivity values for urban surfaces are given as 0.95-0.97 but in Table 2 much lower values of 0.85, 0.9 and 0.95 are given, which seems to introduce an inconsistency. How would an emissivity of 0.85 affect the uncertainty estimation in L7?

- P17 L15-17: Define $f_{dir}$ here as it does not seem to be defined in the main text
- P17 Fig 5 caption: The letters referring to the various panels needs correcting here and in the main text.
- P17 Fig 5 caption and in other captions: Personally I find it quite confusing to keep reading about the 5 days that were discarded for spin-up. Why not define a simulation period and a study period/analysis period (27 June-9 July) in Section 3.1, and then refer to this study period in the figure and table captions? Repetitions between captions could also be reduced.
- P19 Table 4 caption: Some inconsistency with the symbol for temperature that needs to be fixed (this was $T$ previously, and is now $T_{air}$)
- P19 L6: Suggest changing 'a small increase' to 'a very small increase' and presumably 'relative humidity' should be 'specific humidity' seeing as the units given are g kg$^{-1}$
- P19 L10: The wind speed overestimation occurs often: suggest changing 'a few days' to 'most days'
- P19 L9-12: The modelled wind speed is large, often exceeding 5 m s$^{-1}$. This seems quite large for a thermally driven circulation between city and surroundings. Perhaps the prescribed roughness lengths could also be a relevant factor. Did the authors consider this and could they comment on the impact of uncertainty in the prescribed roughness lengths on the evaluation results?
- P20 L21: 'may be related to exposure of the sensor to local influences' – please explain
- P21 Fig 6: Why are no results for BLER plotted here? This would help the reader to interpret Fig 7.
- P21 L4-7: A little deeper discussion would be helpful here. The performance of the model seems to go in the wrong direction in terms of comparison against the observations but this is because of the dominating effect of the temperature bias. However, the newly-incorporated processes seem to cause a change in the expected direction. This cautions against tuning models to match observations
- P22 L10-12: This two-sentence paragraph should be extended to include more discussion (with references from the literature) about the uncertainties of satellite-derived LST, particularly in urban areas. In Fig 8, do you trust the spatial patterns seen in the satellite observations? Crawford et al- (2018) discuss some of the biases affecting LST measurements in urban areas and how these vary spatially.
  Crawford B, Grimmond CSB, Gabey A, Marconcini M, Ward HC, Kent CW (2018) Variability of urban surface temperatures and implications for aerodynamic energy exchange in unstable conditions, QJRMS 144: 1719–1741 DOI: 10.1002/qj.3325
- P26 L3-4: Can the authors explain the finding that both increasing and decreasing leaf area density leads to a reduction in night-time temperature?

Minor points and suggestions for improving language and readability

- P2 L7: interactions concerning moisture should also be listed here
- P2 L11: 'heat interactions' is probably not the right term here, suggest changing to 'latent heat' or 'evaporation'
- P2 L12: 'excerpts' → 'extracts'
- P3 L4: 'Several… …fabrics' – this sentence is redundant given the previous and following sentences so can be deleted
- P3 L7: 'regarded' → 'concerns'
- P3 L9: 'regards' → 'concerns'
- P3 L11-12 and throughout: I would use the singular, i.e. 'street tree datasets' instead of 'street trees datasets'; 'street tree characteristics' instead of 'street trees characteristics' (otherwise you probably need an apostrophe)

- P3 L16: Suggest deleting 'in the presence of street trees'
- P3 L26: Suggest deleting 'the modelling' and adding to the end of this sentence 'by more realistic representation of the surface and its interactions', or similar
- P4 L4: 'garden' → 'gardens'
- P4 L6-8: There is no mention of Section 4 here which seems odd
- P4 L12, L24: It is not clear that '(COSMO)' is a reference. You could delete it from L12 and perhaps provide a year or give the URL in L24 to avoid confusion.
- P4 L20-22: The items in this list are not strictly all processes, suggest rephrasing.
- P5 Fig1 caption: 'the Sections' → 'Sections'
- P5 L6: Delete 'top of'
- P6 L4: Delete the second instance of 'i' as it seems as though 'the ray i' is defined twice
- P6 L25: Seems strange to define the density here before $S_q$
- P7 L3: 'latter' → 'canyon geometry inputs' and 'includes' → 'include'
- P7 L16: 'a measurement site' → 'measurements'
- P7 L23-5: This sentence disrupts the flow here. Suggest deleting or making this the first sentence of Section 2.3.
- P7 L26: Would 'proportions' be better than 'configurations' here? – the model does not account for the arrangement of the surfaces or whether there is one larger urban surface or many smaller urban surfaces with the same total area
- P8 Fig 2 caption: 'Rose' → 'Orange'. Also most of these are quantities not 'operations' – suggest rephrasing
- P8 L1: Delete 'street canyon' (as sky is not really part of the street canyon)
- P10 L4: Start this sentence with 'Following Schubert…'
- P10 L13: 'UPC' → 'UCP'
- P10 L14: The reference to Fig 2 would be more helpful in L16
- P11 L1-8: Change the numbers in this section so that commas rather than apostrophes separate the digits (e.g. '830'000' → '830,000')
- P12 Fig 3 caption: State what the dashed line represents in the figure caption
- P13 L1: 'in the centre-south and centre-west' → 'south and west of the centre'
- P14 Fig 4 caption: 'urban tile' is probably be better than 'urban canopy' as the latter could be interpreted as trees above buildings
- P14 Fig4 caption: Change to 'Further details about the variables are given in Table A1'
- P14 Table 2 and in other tables too: Use a consistent number of decimal places, e.g. 0.10 0.10 and 0.15 for albedo; 0.85, 0.90 and 0.95 for emissivity
- P15 L4: Change to 'vehicles and seasonal'
- P15 L7: It would be more useful to explicitly state the period over which the average was calculated
- P15 L13 and elsewhere: Should this be 'MCR-Lab' not 'MRC-Lab'?
- P15 L14: 'building' → 'buildings'
- P15 L17: ', but' → ' and'
- P15 L17: 'estimated to' → 'estimated at' or 'estimated to be'
- P15 L22: You could add 'in general' to make it clear that these considerable uncertainties are not only for the Basel dataset
- P16 L7: 'placed' → 'located'
- P16 L8: 'then' → 'than'
- P16 L8-9: 'in the area' – in which area? The urban canopy layer is extremely spatially variable so there would be considerable temperature variation expected throughout any 'area'
- P16 L19: Delete 'as well'
- P16 L21: Change to 'Given the vegetated environment and distance from the city centre,' to avoid misinterpretation
- P16 L21-22: Change to 'as a rural reference'
- P17 L30: Delete 'will'

- P17 Fig5: Delete the unhelpful minor tick marks on the right-hand plots.
- P19 Table 4 caption: Change '(see Table 1… set-up)' to '(LA0)'
- P19 Table4 caption: You could point out that the units only apply to RMSE and MBE, not $r^2$
- P19 L11: Change 'and' to 'or'
- P19 L15: Change 'of' to 'by'
- P20 L3: Shouldn't '0.06' be '0.04'?
- P20 L6: Suggest deleting 'slight' as the overestimation is appreciable. Also add that $r^2$ gets worse here.
- P20 L16: Delete '2-m air temperature'
- P20 L17: Delete 'will'
- P20 L30-31: Avoid this two-sentence paragraph by moving 'The statistical scores…only.' to the end of the previous paragraph and 'The comparison between… Fig. 7.' to the start of the following paragraph.
- P21 L1: 'lower' → 'lesser'
- P21 L14 'outskirt' → 'outskirts'
- P22 L3: Change 'only quite' to 'quite'
- P22 L3: Change to '…STD and LAO are seen, with STD…'
- P23 L2: 'ST' → 'surface temperature'
- P23 L6: Why is 2-m air temperature now denoted $T_{air}$ and not $T_{2m}$ as before?
- P24 Table 7: Units of surface temperature are missing
- P26 Fig9 caption: Change '$S$' to '$S_\varphi$' in line with Eq 15
- P26 L13-4: Join this first sentence with the following paragraph
- P27 L3: Change to 'of the clumping index'
- P27 L6-7: Change to 'the temporal changes of $\Omega$ with solar zenith angle'
- P27 L24: Change to 'values of the material properties for roof, wall and street elements were used'
- P27 L28: Should read 'WUDAPT'
- P28 L15: Change to: 'generally well, although its magnitude'
- P28 L16: Change 'it' to 'this difference'
- P28 L18-21: Again some mention of uncertainties in LST would be helpful here
- P28 L23: Change to 'found only at some'
- P29 L7: Close bracket missing after the email address
- P29 L9: Change to 'as a standalone'
- P30 L12-3: Define $A_F$
- P31 L1: 'heigh' → 'height'
- P32 L24 and L32: Check reference formatting

---

## Referee Comment (RC2) · Anonymous Referee #2 · 21 Oct 2019

This paper describes and evaluates COSMO-BEP-Tree v1.0: a coupled urban climate model with explicit representation of street trees. The authors assess the model during a heatwave event with observations from flux towers, surface stations and satellites.

The authors have presented a very strong study which is thorough and convincing. The issue of integrating the effects of street trees in urban meteorological models is important because increasing tree cover is "go-to" response of urban planners who wish to reduce urban heat impacts. As the manuscript shows, interacting processes are complex and outcomes not always obvious. The evaluation is clear and thorough, and the technical achievements will be beneficial for future urban studies.

[Figure]

The lack of open availability of the source code is disappointing for a study presented in GMD. The process for accessing code outlined in the study is not timely, and includes mailing physical documents and/or negotiation. I have therefore not reviewed the code. The authors should reconsider publishing the code openly on a persistent public archive.

Detailed comments follow.

Pg 2 ln 12: "Tree transpiration reduces the surface temperature of the foliage by converting part of the solar radiation to latent heat" not only solar radiation.

Pg 2 ln 15: "Modelling studies on the cooling potential of street trees have almost exclusively been performed at the scale of individual street canyons to single neighbourhood... Only very few studies investigate the city-wide impact of street trees, mainly due to the limited availability of models able to represent street trees at the scale of the city..."

It would be useful here to acknowledge the various mesoscale urban meterological models which have incorporated vegetation as low-height gardens within the street canyons (e.g. Thatcher and Hurley, 2012; Lemonsu et al, 2012; Wang et al 2013). Although not "street-trees" (as they do not provide shading on walls or reduce canyon skyview), they do shade the ground and interact directly with canyon radiation/ turbulent fluxes and momentum budgets, alter the Bowen ratio and reduce canyon surface temperatures. This will also give an opportunity to be clearer about the authors definition of "street trees" when the concept is introduced, as readers may assume models with in-canyon low vegetation, or even external vegetation tiles, are doing the same thing. The statement "very few studies investigate the city-wide impact of street trees" could then be more carefully stated, as there have been many studies at city or larger scales which have assessed the impact of urban vegetation, but most have used schemes with low vegetation or used a tiled approach, and hence missed important shading and skyview effect which is the strength of the current study.

[Figure]

* Thatcher M and Hurley P 2012 Simulating Australian urban climate in a mesoscale atmospheric numerical model Boundary-Layer Meteorol 142 149–75

* Lemonsu A, Masson V, Shashua-Bar L, Erell E and Pearlmutter D 2012 Inclusion of vegetation in the Town Energy Balance model for modelling urban green areas Geosci. Model Dev. 5 1377–93

* Wang Z-H, Bou-Zeid E and Smith J A 2013 A coupled energy transport and hydrological model for urban canopies evaluated using a wireless sensor network Quarterly Journal of the Royal Meteorological Society 139 1643–57

pg 6 ln 4: Why are the intensity terms ($\Delta V\_i$ and $r\_i$) dimensionless? Shouldn't they have units?

pg 8 ln 13 "street trees do not interact with soil moisture content as represented by COSMOS's land surface scheme" More explanation as to why this was the case – technical constraints? Or was it assumed that all soil is constantly irrigated in urban areas? You may wish to mention the drawback of this assumption (i.e. overestimating latent heat) at this time. Otherwise, do the underlying LSM soil and urban scheme exchange any fluxes? How? Figure 1 could be used here to better explain how fluxes of the urban scheme are coupled to the atmosphere and LSM.

pg 10 ln 6: For the last terms of eq 13, why is Ts associated with fnat, and Tg be with furb? I don't understand whether this temperature averaging approach is valid. I can see the equation is taken from Schubert (2013), but it is not explained there either. For example, take the extreme position where fnat=0, furb=1, then the equation simplifies to:

T2m = Ts + r (T1 – Tg)

Why is Tg part of the T2m temperature if there is no fnat? Also, how is the urban surface temperature here defined? Does it, for example, include the roof facets? I was pleased to see the clear definition of surface temperature for the evaluation of the satel-

lite observations (pg 17) which was based on the facets that the satellite sees (roofs streets and natural surfaces rather than walls). But it's not clear that definition is appropriate for 2m temperature out of sight of roofs, and within sight of walls. If the same satellite-based definition is used for the T2m calculation, authors should note that's likely to lead to underestimation of the cooling benefit of street trees for temperatures within the canyon.

pg 11 ln 3: It would be useful to include information here about the average fraction of green cover in Basel, for easy comparison with other urban studies.

p 11 ln 23: There is discussion of atmospheric initial conditions and on soil properties but it is not clear how soil moisture was initialised. Soil moisture has significant impact on the intensity of heatwaves (e.g. see Wang et al 2019), and soil moisture has memory much longer than the 5-day spinup, so this variable needs to be carefully initialised.

* Wang P, Zhang Q, Yang Y and Tang J 2019 The Sensitivity to Initial Soil Moisture for Three Severe Cases of Heat Waves Over Eastern China Front. Environ. Sci. 7

pg 13 Table 1, please provide explanation for name acronyms here for easier reference.

pg 18 ln 4. I didn't find the attribution of the night-time overestimation of air temperature to the inability of COSMO to represent nocturnal stable boundary layer conditions convincing. If it were an issue with COSMO failing to represent a stable boundary layer, then it should occur in the non-urban site (BLER) night-time cooling rate. However, the supplementary Figure 1 indicates the rate of cooling matches observations well (although with a positive bias throughout), which to me indicates the evolution of stability conditions are probably reasonable at the rural site. It therefore seems the overestimation of night-time urban air temperatures could just as easily be from issues within the urban scheme configuration rather than the atmospheric model or stability issues. The supplementary Figure 2 indicates the bulk albedo may be low, and the material properties of facets listed in Table 2 store a lot of heat. I understand those values are based on previous studies, but those values were derived through optimising WRF SLUCM,

and so are not necessarily realistic urban parameters, but could simply be accounting for deficiencies in that model. Therefore, the parameters won't necessarily be appropriate for BEP-TREE at Basel. I'm not asking for the simulations to be redone, but the authors shouldn't be so quick to simply attribute night-time errors to the atmospheric model when urban parameters may be the issue (low albedo, high heat storage).

pg 20 ln 27 "are" -> area

pg 26 "Future Work" It was noted earlier that BEP-TREE does not interact with LSM soil moisture - this is a major limitation which should be discussed here.

pg 27 ln 21 "attributed to the inability of the model to reproduce a very stable atmospheric conditions" again, if this argument is used it should be better supported with information from the current study. For example, was the minimum value of turbulent diffusion coefficients (K_i,min) set at 1 m2/s (per Buzzi 2011) in the current experiments? What were the stability conditions at the rural site? Did an inversion form? Why is the rate of cooling at night at the rural site seemingly correct if a stability is incorrectly simulated? Do the observed wind speeds support the conclusion that very stable atmospheric conditions should have formed on those nights? Is there other observational support for the stability or non-stability of rural sites in the area (non-urban flux towers, boundary layer measurements etc)? Finally, I found myself referring to the rural figure often throughout the manuscript, it may be better placed in the main body near figure 6.

pg 28 ln 16 "additionally explained by an underestimation of the night-time temperature" I believe the night-time temperature is overestimated at the BKLI site.
* * *

---

## Author Comment (AC1) · 16 Jan 2020

**Reply to reviewer's comments**

Typographical convention:

Black -> reviewer's comment

Blue -> author's comment

Red -> modification to the manuscript

Underlined -> text added to the original version

 -> text removed from the original version

The line numbers refer to the original version of the manuscript.

**Reviewer 1 (RC1)**

Realistic representation of urban vegetation is an important step to improving model capabilities and performance. In this paper, a vegetated urban canopy model incorporating street trees (BEP- Tree) is coupled with a mesoscale model (COSMO). The authors briefly describe the relevant parts of the models and their combination, before evaluating the combined model against a range of observations. The work is well-presented, with a clear structure and, mostly, an appropriate level of detail (although in some places more discussion would be beneficial). The inclusion of street trees within the model is shown to make generally small improvements to the meteorology but substantial improvements to the sensible and latent heat fluxes. The coupled model will be useful for applications concerning urban greening as well as for more general studies which will benefit from better representation of vegetation processes.

The paper is very well-presented overall. There are a few consistency issues which need to be addressed and a little more explanation would be beneficial in parts. However, I recommend this publication following minor revisions.

We are happy that the reviewer appreciated the paper. We included further discussion and fixed the inconsistencies.

Main points

- P8 L15: The lack of interaction with soil moisture seems to be the major weakness in this study. More discussion would be beneficial. Why was the decision made not to implement this interaction? Can the authors comment on the effect of this decision? How realistic is it to assume transpiration in urban areas is never limited by soil moisture (I would think not very realistic)?

We added a discussion on the lack of interaction with soil moisture.

P8 L15 "Although moisture exchange between street trees and the atmosphere is implemented, street trees do not interact with soil moisture content as represented by COSMO's land surface scheme. In other words, a mechanistic interaction between soil moisture and the transpiration of street trees is

not included, assuming that the transpiration is never limited by soil water availability. A careful representation of soil moisture in the urban tiles would have required the development of a new urban hydrology scheme, which was beyond the scope of the study. The missing interaction with soil moisture may reduce the model ability to represent variations in transpiration during periods with large changes in soil moisture (Konarska et al., 2016; Asawa et al., 2017). Nevertheless, street trees are less sensitive to variations in soil moisture than short vegetation, thanks to their deeper root system (Chen et al., 2011; Asawa et al., 2017)."

Additionally, the section "Future studies" is extended accordingly.

P26, L18 "In order to improve the representation of the transpiration from street trees and therefore the modelling of the associated latent heat fluxes, a mechanistic stomata model (e.g. Damour et al., 2010) should be implemented. Furthermore, to properly represent soil water availability during extended drought periods, an urban hydrology module should be developed and coupled to COSMO-BEP-Tree (e.g. Järvi et al., 2011, Yang et al., 2015, Stavropulos-Laffaille et al., 2018)."

Additional references:

Chen, Lixin, et al. "Biophysical control of whole tree transpiration under an urban environment in Northern China." *Journal of Hydrology* 402.3-4 (2011): 388-400. https://doi.org/10.1016/j.jhydrol.2011.03.034

Asawa, Takashi, Tomoki Kiyono, and Akira Hoyano. "Continuous measurement of whole-tree water balance for studying urban tree transpiration." *Hydrological Processes* 31.17 (2017): 3056-3068. https://doi.org/10.1002/hyp.11244

Järvi, Leena, C. S. B. Grimmond, and Andreas Christen. "The surface urban energy and water balance scheme (SUEWS): Evaluation in Los Angeles and Vancouver." *Journal of hydrology* 411.3-4 (2011): 219-237. https://doi.org/10.1016/j.jhydrol.2011.10.001

Yang, Jiachuan, et al. "Enhancing hydrologic modelling in the coupled weather research and forecasting–urban modelling system." *Boundary-Layer Meteorology* 155.1 (2015): 87-109. https://doi.org/10.1007/s10546-014-9991-6

Stavropulos-Laffaille, Xenia, et al. "Improvements to the hydrological processes of the Town Energy Balance model (TEB-Veg, SURFEX v7. 3) for urban modelling and impact assessment." Geoscientific Model Development 11.10 (2018): 4175-4194. https://doi.org/10.5194/gmd-11-4175-2018

Konarska, Janina et al. "Transpiration of urban trees and its cooling effect in a high latitude city." International journal of biometeorology 60 (2016): 159–172. https://doi.org/10.1007/s00484-015-1014-x

- P10 L6: It is not clear how $T_g$ relates to $T_{nat}$ or $T_s$ relates to $T_{urb}$

We apologize that there was a typo in the second part of the equation. $f_{nat}$ should always go with $T_g$ and $f_{urb}$ with $T_s$ . Additionally, the definition of $T_s$ was not consistent with its use later in the manuscript (where it is referred to as $T_{str}$).

We corrected the equation and the following explanation.

P10, L6 "$T_{2m} = (f_{nat}\ T_g + f_{urb}\ \cancel{T_s}\underline{T_{str}}) + r\ (T_1 - (f_{nat}\ \cancel{T_s}\underline{T_g} + f_{urb}\ \cancel{T_g}\underline{T_{str}}))$,

where $T_g$ is the surface temperature of the natural tile, $T_{str}$ is the street surface temperature , … "

- P11 L5: Provide some justification for the statement about Basel's 'abundant green infrastructures' as the statistics and Fig 3b do not suggest a particularly vegetated city.

This statement was based on information provided by the city offices. We reviewed the scientific literature but did not find any studies that substantiate our statement. Therefore, we rephrased the sentence and removed the statement about 'abundant green infrastructures'.

In order to allow a comparison with other studies, we calculated the average canopy cover from street trees and added an according sentence to the manuscript. A figure with the variability of the canopy cover from street trees in the different districts is added to the Supplement (Figure R1).

P11, L3: "The inner city (Basel-Stadt) includes  more than 24'000 urban trees and 275 hectares of public vegetated surfaces distributed over a total area of the city centre of approx. 2'385 hectares (Stadtgaertnerei Kantons Basel-Stadt). The average canopy cover from street trees is about 20% when considering the entire urban area, but it varies from 6 to 31% in the different districts (see Supplement). The city of Basel is  an interesting target for  the availability of extensive observational data sets for model evaluation, which had been used in numerous previous urban climate studies (e.g. Rotach et al., 2005; Parlow et al., 2014)."

[Figure]

**Figure R1 - Canopy cover from street trees in the administrative districts of Basel-Stadt. The building geometries are shown in the background.**

- P11 L16: Can the authors add whether any spurious structures occurred (as in other studies, e.g. Salamanca et al. 2012) given the small horizontal grid spacing used is in the 'grey zone'? Salamanca F, Martilli A, Yague C (2012) A numerical study of the urban heat island over Madrid during the DESIREX (2008) field campaign with WRF and an evaluation of simple mitigation strategies, IJC 32: 2372–2386 DOI: 10.1002/joc.3398

Clear spurious structures were not detected in the simulations.

Nevertheless, a sentence regarding the use of the model in the so-called grey-zone is added in the section "future studies".

P27, L28: "It has to be noted that the model resolution in this study falls within the so-called "grey zone" or "terra incognita" (Wyngaard, 2004) of meso-scale atmospheric modelling, a condition where the horizontal grid spacing and hence the scale of the spatial filter used on the equations of motion are of the same order as the scale of the energy- and flux-containing turbulence. This condition may produce a misrepresentation in the turbulence and wind speed patterns (Efstathiou et al., 2015). High-resolution urban climate studies reported the formation of horizontal convective rolls which have an unclear counterpart in the real atmosphere (Salamanca et al., 2012; Gutiérrez et al., 2015). Although such issues were not visible in this application, these aspects need to be considered in future studies, for example by exploring the performance of new scale-aware turbulence schemes (e.g. Junshi et al., 2015, Shin et al., 2015) or large-eddy-simulation schemes as recently developed for the COSMO model (Panosetti et al., 2016)."

Additional references:

Efstathiou, G. A., and Robert J. Beare. "Quantifying and improving sub-grid diffusion in the boundary-layer grey zone." *Quarterly Journal of the Royal Meteorological Society* 141.693 (2015): 3006-3017. https://doi.org/10.1002/qj.2585

Gutiérrez, Estatio, et al. "Simulations of a heat-wave event in New York City using a multilayer urban parameterization." *Journal of Applied Meteorology and Climatology* 54.2 (2015): 283-301. https://doi.org/10.1175/JAMC-D-14-0028.1

Shin, Hyeyum Hailey, and Song-You Hong. "Representation of the subgrid-scale turbulent transport in convective boundary layers at gray-zone resolutions." *Monthly Weather Review* 143.1 (2015): 250-271. https://doi.org/10.1175/MWR-D-14-00116.1

Ito, Junshi, et al. "An extension of the Mellor–Yamada model to the terra incognita zone for dry convective mixed layers in the free convection regime." *Boundary-layer meteorology* 157.1 (2015): 23-43. https://doi.org/10.1007/s10546-015-0045-5

Panosetti, Davide, et al. "Idealized large-eddy and convection-resolving simulations of moist convection over mountainous terrain." *Journal of the Atmospheric Sciences* 73.10 (2016): 4021-4041. https://doi.org/10.1175/JAS-D-15-0341.1

Salamanca, Francisco, Alberto Martilli, and Carlos Yagüe. "A numerical study of the Urban Heat Island over Madrid during the DESIREX (2008) campaign with WRF and an evaluation of simple mitigation strategies." International Journal of Climatology 32.15 (2012): 2372-2386. https://doi.org/10.1002/joc.3398

Wyngaard, John C. "Toward numerical modeling in the "Terra Incognita"." *Journal of the atmospheric sciences* 61.14 (2004): 1816-1826. https://doi.org/10.1175/1520-0469(2004)061<1816:TNMITT>2.0.CO;2

- P12 Fig 3: Station WEIL does not seem to be used anywhere in the manuscript – delete from the map

The station WEIL is removed from Figure 3.

- P13 L3-4: Two sentences providing a general description of the imperviousness dataset and the buildings dataset would be helpful (i.e. are these fractions of impervious surface cover or degree of imperviousness; is it building height or building material or building use; at what resolution?)

We added two sentences describing the imperviousness dataset and the building dataset.

P13, L5: "The urban canopy parameters were derived with UCPgenerator from the following input datasets: (a) imperviousness from EEA (2015), (b) 3D buildings from Federal Office of Topography (2007) and (c) lidar-based tree canopy height (see data availability section). The imperviousness dataset represents the percentage of soil sealing (including building area) and has a resolution of 20 m. The building dataset is a vector dataset representing the building geometries (accuracy 3-8 m), including heights but without information on the building materials or use."

- P15 Table 3: Latent heat flux seems to be denoted $Q_L$ here and $Q_E$ later in the manuscript – make consistent in all text, tables and figures. Fluxes also appear in the row for BLER – presumably this is incorrect?

Thank you for pointing out this. We removed the inconsistency in latent heat flux notation throughout the manuscript. We removed the fluxes from Table 3.

- P15 L21-22: 'net' → 'four-component' as the CNR4 measures all four radiation components individually. Also, the radiation measurements are mentioned here but not used in the evaluation (e.g. in Fig 5). Why not? Can the authors comment on the results for the radiation components for STD and LA0 runs?

Thank you for pointing this out. We corrected the description of the CNR4 measurement.

P15, L21-22: "radiation fluxes are measured with a  four-component radiometer"

The comparison with radiation is only shown in the Supplement, since we expected only a minor impact

from street trees.
We added the following explanation regarding the evaluation in terms of radiation fluxes:

P20, L12: "For completeness, the comparison between modelled and observed radiation fluxes is shown in the Supplement. The model simulations agree reasonably well with the observed radiation fluxes, although the upward short-wave radiation is underestimated. This indicates that the albedo is underestimated, which can be related to the choice of material properties (Table 2). The upward long-wave radiation is slightly overestimated, which can therefore be related to the underestimation of the albedo and, consequently, to an overestimation of surface temperatures. Overall, the representation of street trees had a very minor impact on the radiation fluxes."

- P16 L4: Aren't there only five surface stations (depending on whether the WEIL data should be included or not)? Can the authors clarify here why no results from WEIL have been included?

Thank you for pointing out this inconsistency.
There are indeed only five sites since the site of WEIL has not been used.

P16, L4: " Five surface stations are used …"

- P17 L5: Here the emissivity values for urban surfaces are given as 0.95-0.97 but in Table 2 much lower values of 0.85, 0.9 and 0.95 are given, which seems to introduce an inconsistency. How would an emissivity of 0.85 affect the uncertainty estimation in L7?

Thank you for pointing out this inconsistency.

We recalculated the LST with the values of emissivity as used in the model from Loridan and Grimmond (2012) and compared with those recommended by Sobrino et al., (2008) and Baldridge et al., (2009). We added an additional figure in the Supplement (Figure R1) that highlights the impact of this choice on the LST and edited the text with a comment on the uncertainty introduced by this aspect.

[Figure]

**Figure R2 – Impact of the choice of emissivity values for urban surfaces on the land surface temperature (LST). (a,b) show the observed LST with emissivity values from Sobrino et al., (2008) and Loridan and Grimmond (2012), respectively; (c) shows the difference in LST between the two approaches.**

"A modified vegetation-threshold approach by Sobrino et al., (2008) was used to estimate the surface emissivity. As the original approach does not account for built-up areas, a lookup table for dark and bright urban surfaces with a range of 0.95 to 0.97, based on emissivity values for typical urban fabrics (Baldridge et al., 2009), was applied (Mitraka et al., 2012). Note that these values differ from those used in the model (Table 2). The uncertainty in the values of emissivity produces average changes in the LST of about 1.25 K (Supplement). On top of this, the LST is affected by uncertainties due to atmospheric correction, sensor and off-nadir view (Voogt and Oke, 2003; Chen et al., 2017) "

- P17 L15-17: Define $f_{dir}$ here as it does not seem to be defined in the main text

We added a definition of $f_{dir}$ .

P17, L15-16: "…, $f_{dir}$ is the fraction of canyons with direction $d$, …"

- P17 Fig 5 caption: The letters referring to the various panels needs correcting here and in the main text.

P18, Fig5 caption: "Comparison between observations (Obs) and model simulations (STD, LA0) of air temperature (a,b), specific humidity (c,d), wind speed (e,f), sensible heat flux (g,h) and latent heat flux (i,l) at the site BKLI (38 m tall flux tower) during the selected simulation period (discarding the first 5 days as spin-up). The left (a,c,e,g,i) and right (b,d,f,h,l) columns show the 3-h moving average and the diurnal cycle, respectively. …"

P18, L1-2: "The model simulates the evolution of air temperature very well during the evaluation period (Figure 5a-b) …"

P19, L1: "The evolution of specific humidity $q$ throughout the simulation is generally well reproduced (Figure 5c-d)."

P19, L9: "The model simulates well the evolution of wind speed ($u$) during the evaluation period (Figure 5 e-f), …"

P19, L16: "The sensible heat flux ($Q_H$) follows very well the observations during the evaluation period (Figure 5g-h), …"

P20, L4: "The evolution of latent heat flux ($Q_E$) is generally well reproduced by the model (Figure 5-l), …"

- P17 Fig 5 caption and in other captions: Personally I find it quite confusing to keep reading about

the 5 days that were discarded for spin-up. Why not define a simulation period and a study period/analysis period (27 June-9 July) in Section 3.1, and then refer to this study period in the figure and table captions? Repetitions between captions could also be reduced.

Following this suggestion, we applied the following changes to the manuscript:

P11, L9: "In this study, COSMO-BEP-Tree was run for the period 22 June 2015 - 9 July 2015, with the first 5 days discarded as spin-up period. The analyses were performed on the period 27 June 2015 - 9 July 2015, called study period hereinafter."

P18, Figure 5 caption: "… during the  study period . …"

P18, Table 4 caption: "… during the study period . …"

P21, Figure 6 caption: "…during the  study period . …"

P22, Table 5 caption: "during the study period . …"

P24, Table 6 caption: "during the study period . …"

- P19 Table 4 caption: Some inconsistency with the symbol for temperature that needs to be fixed (this was $T$ previously, and is now $T_{air}$)

Table 4 is updated with $T$ instead of $T_{air}$.

- P19 L6: Suggest changing 'a small increase' to 'a very small increase' and presumably 'relative humidity' should be 'specific humidity' seeing as the units given are g kg$^{-1}$

We agree with the reviewer and changed the text accordingly.

- P19 L10: The wind speed overestimation occurs often: suggest changing 'a few days' to 'most days'

We agree with the reviewer and changed the text accordingly.

- P19 L9-12: The modelled wind speed is large, often exceeding 5 m s$^{-1}$. This seems quite large for a thermally driven circulation between city and surroundings. Perhaps the prescribed roughness lengths could also be a relevant factor. Did the authors consider this and could they comment on the impact of uncertainty in the prescribed roughness lengths on the evaluation results?

First of all, we add the prescribed values of roughness length to Table 2. The values correspond to the

default options of the urban canopy model BEP, as used in previous studies (Chen et al., 2011; Salamanca et al., 2011; Hammerberg et al., 2018).

| | Roof | Wall | Road |
|---|---|---|---|
| Albedo (-) | 0.1 | 0.1 | 0.15 |
| Emissivity (-) | 0.85 | 0.9 | 0.95 |
| Heat Capacity (J m−3 K−1 × 106) | 1.5 | 1.4 | 1.5 |
| Heat Conductivity (W m−1 K−1) | 0.8 | 1.0 | 0.8 |
| Total thickness (m) | 0.5 | 0.3 | 1.0 |
| Roughness length (m) | 0.01 | - | 0.01 |

Table 2. Material properties used in the urban canopy model.

We agree with the reviewer that wind speed may be sensitive to values of roughness length at individual surfaces. However, we haven't performed any sensitivity studies on the impact of the prescribed values of roughness length. Moreover, we are also not aware of any studies in the literature where this aspect has been investigated in the context of mesoscale modelling with multi-layer urban canopy model.

Nevertheless, a similar overestimation is also observed at the rural site (Figure 1 in the Supplement). In our opinion, this points out to an overestimation of regional scale wind speed rather than local effects in the urban area.

P19, L10: "…The  overestimation was not limited to the urban area but was also found at the rural site (Supplement). Although its origin is unclear, possible causes may be a misrepresentation of the  regional scale wind from the driving boundary conditions and an overestimation of the thermally driven flow induced by the surface temperature gradient between the city and the surroundings."

- P20 L21: 'may be related to exposure of the sensor to local influences' – please explain

We extended the explanation regarding the sensor exposure, as already done when describing the sites (P16, L8).

P19, L10: "This difference may be related to the local exposure of BFEL sensor, located in the vicinity of a (warm) south facing wall."

- P21 Fig 6: Why are no results for BLER plotted here? This would help the reader to interpret Fig 7.

We included the daily profile of air temperature at BLER in Figure 6.

- P21 L4-7: A little deeper discussion would be helpful here. The performance of the model seems to go in the wrong direction in terms of comparison against the observations but this is because of the dominating effect of the temperature bias. However, the newly-incorporated processes seem to cause a change in the expected direction. This cautions against tuning models to match

observations

We included a discussion in order to make it clearer that other model biases were masking the benefit of representing street trees for UHI estimation.

P21 L4-7 "The representation of street trees affects the simulated UHI intensity by producing a slight reduction in UHI intensity from late morning to early evening. The impact of street trees is more evident at sites with higher urban fraction and density of street trees. An improvement in model performance with street trees was, however, not evident in terms of UHI representation. Considering that street trees produce changes in the expected direction, this suggests that probably other model biases are masking the benefit of including street trees. "

- P22 L10-12: This two-sentence paragraph should be extended to include more discussion (with references from the literature) about the uncertainties of satellite-derived LST, particularly in urban areas. In Fig 8, do you trust the spatial patterns seen in the satellite observations? Crawford et al- (2018) discuss some of the biases affecting LST measurements in urban areas and how these vary spatially.
  Crawford B, Grimmond CSB, Gabey A, Marconcini M, Ward HC, Kent CW (2018) Variability of urban surface temperatures and implications for aerodynamic energy exchange in unstable conditions, QJRMS 144: 1719–1741 DOI: 10.1002/qj.3325

We thank the reviewer for the suggestion and extended the paragraph with more discussion about uncertainties in satellite-derived LST in urban areas.

P22 L10-12 "The explicit representation of street trees did not improve the agreement with satellite observations of land surface temperature. However, this observation should be taken with care given that satellite-derived LST in urban areas are known to have a large degree of uncertainly due to potential errors in the surface emissivity, sensor and satellite view bias (Voogt and Oke, 2003; Chen et al., 2017). Slight off-nadir view angles can introduce a bias over areas with tall buildings while LST variations at scales below 500 m may be unresolved (Crawford et al., 2018). "

References:

Voogt, James A., and Tim R. Oke. "Thermal remote sensing of urban climates." Remote sensing of environment 86.3 (2003): 370-384. https://doi.org/10.1016/S0034-4257(03)00079-8

Chen, Feng, et al. "Challenges to quantitative applications of Landsat observations for the urban thermal environment." *Journal of Environmental Sciences* 59 (2017): 80-88. https://doi.org/10.1016/j.jes.2017.02.009

Crawford, Ben, et al. "Variability of urban surface temperatures and implications for aerodynamic energy exchange in unstable conditions." Quarterly Journal of the Royal Meteorological Society 144.715 (2018): 1719-1741. https://doi.org/10.1002/qj.3325

- P26 L3-4: Can the authors explain the finding that both increasing and decreasing leaf area density leads to a reduction in night-time temperature?

We thank the reviewer for the suggestion and included an explanation.

P26, L3-4: "A reduction in the leaf area density (LA-) generally produced an opposite response to that of an increase in leaf area density (LA+). An interesting exception is $T_{air}$ at night, suggesting that different processes with opposing sensitivities to values of LAD are at work. The largest night-time reduction in $T_{air}$ was achieved with LA-, which was associated with a similar reduction in $T_{str}$. A slight reduction in $T_{air}$, however, was also found for LA+, which contrasts with an increase in $T_{str}$ for this scenario. This points towards complex, non-linear interactions between the different effects of LAD on radiation and winds. However, further studies are needed to corroborate this finding."

Minor points and suggestions for improving language and readability
- P2 L7: interactions concerning moisture should also be listed here
- P2 L11: 'heat interactions' is probably not the right term here, suggest changing to 'latent heat' or 'evaporation'
- P2 L12: 'excerpts' → 'extracts'
- P3 L4: 'Several… …fabrics' – this sentence is redundant given the previous and following sentences so can be deleted
- P3 L7: 'regarded' → 'concerns'
- P3 L9: 'regards' → 'concerns'
- P3 L11-12 and throughout: I would use the singular, i.e. 'street tree datasets' instead of 'street trees datasets'; 'street tree characteristics' instead of 'street trees characteristics' (otherwise you probably need an apostrophe)
- P3 L16: Suggest deleting 'in the presence of street trees'
- P3 L26: Suggest deleting 'the modelling' and adding to the end of this sentence 'by more realistic representation of the surface and its interactions', or similar
- P4 L4: 'garden' → 'gardens'
- P4 L6-8: There is no mention of Section 4 here which seems odd
- P4 L12, L24: It is not clear that '(COSMO)' is a reference. You could delete it from L12 and perhaps provide a year or give the URL in L24 to avoid confusion.
- P4 L20-22: The items in this list are not strictly all processes, suggest rephrasing.
- P5 Fig1 caption: 'the Sections' → 'Sections'
- P5 L6: Delete 'top of'
- P6 L4: Delete the second instance of 'i' as it seems as though 'the ray i' is defined twice
- P6 L25: Seems strange to define the density here before $S_q$
- P7 L3: 'latter' → 'canyon geometry inputs' and 'includes' → 'include'
- P7 L16: 'a measurement site' → 'measurements'
- P7 L23-5: This sentence disrupts the flow here. Suggest deleting or making this the first sentence of Section 2.3.
- P7 L26: Would 'proportions' be better than 'configurations' here? – the model does not account for the arrangement of the surfaces or whether there is one larger urban surface or many smaller urban surfaces with the same total area
- P8 Fig 2 caption: 'Rose' → 'Orange'. Also most of these are quantities not 'operations' – suggest rephrasing

- P8 L1: Delete 'street canyon' (as sky is not really part of the street canyon)
- P10 L4: Start this sentence with 'Following Schubert…'
- P10 L13: 'UPC' → 'UCP'
- P10 L14: The reference to Fig 2 would be more helpful in L16
- P11 L1-8: Change the numbers in this section so that commas rather than apostrophes separate the digits (e.g. '830'000' → '830,000')
- P12 Fig 3 caption: State what the dashed line represents in the figure caption
- P13 L1: 'in the centre-south and centre-west' → 'south and west of the centre'
- P14 Fig 4 caption: 'urban tile' is probably be better than 'urban canopy' as the latter could be interpreted as trees above buildings
- P14 Fig4 caption: Change to 'Further details about the variables are given in Table A1'
- P14 Table 2 and in other tables too: Use a consistent number of decimal places, e.g. 0.10 0.10 and 0.15 for albedo; 0.85, 0.90 and 0.95 for emissivity
- P15 L4: Change to 'vehicles and seasonal'
- P15 L7: It would be more useful to explicitly state the period over which the average was calculated
- P15 L13 and elsewhere: Should this be 'MCR-Lab' not 'MRC-Lab'?
- P15 L14: 'building' → 'buildings'
- P15 L17: ', but' → ' and'
- P15 L17: 'estimated to' → 'estimated at' or 'estimated to be'
- P15 L22: You could add 'in general' to make it clear that these considerable uncertainties are not only for the Basel dataset
- P16 L7: 'placed' → 'located'
- P16 L8: 'then' → 'than'
- P16 L8-9: 'in the area' – in which area? The urban canopy layer is extremely spatially variable so there would be considerable temperature variation expected throughout any 'area'
- P16 L19: Delete 'as well'
- P16 L21: Change to 'Given the vegetated environment and distance from the city centre,' to avoid misinterpretation
- P16 L21-22: Change to 'as a rural reference'

- P17 L30: Delete 'will'

- P17 Fig5: Delete the unhelpful minor tick marks on the right-hand plots.
- P19 Table 4 caption: Change '(see Table 1… set-up)' to '(LA0)'
- P19 Table4 caption: You could point out that the units only apply to RMSE and MBE, not $r^2$
- P19 L11: Change 'and' to 'or'
- P19 L15: Change 'of' to 'by'
- P20 L3: Shouldn't '0.06' be '0.04'?
- P20 L6: Suggest deleting 'slight' as the overestimation is appreciable. Also add that $r^2$ gets worse here.
- P20 L16: Delete '2-m air temperature'
- P20 L17: Delete 'will'
- P20 L30-31: Avoid this two-sentence paragraph by moving 'The statistical scores…only.' to the end of the previous paragraph and 'The comparison between… Fig. 7.' to the start of the following paragraph.
- P21 L1: 'lower' → 'lesser'
- P21 L14 'outskirt' → 'outskirts'
- P22 L3: Change 'only quite' to 'quite'
- P22 L3: Change to '…STD and LAO are seen, with STD…'
- P23 L2: 'ST' → 'surface temperature'
- P23 L6: Why is 2-m air temperature now denoted $T_{air}$ and not $T_{2m}$ as before?
- P24 Table 7: Units of surface temperature are missing
- P26 Fig9 caption: Change '$S$' to '$S_\varphi$' in line with Eq 15

- P26 L13-4: Join this first sentence with the following paragraph
- P27 L3: Change to 'of the clumping index'
- P27 L6-7: Change to 'the temporal changes of $\Omega$ with solar zenith angle'
- P27 L24: Change to 'values of the material properties for roof, wall and street elements were used'
- P27 L28: Should read 'WUDAPT'
- P28 L15: Change to: 'generally well, although its magnitude'
- P28 L16: Change 'it' to 'this difference'
- P28 L18-21: Again some mention of uncertainties in LST would be helpful here
- P28 L23: Change to 'found only at some'
- P29 L7: Close bracket missing after the email address
- P29 L9: Change to 'as a standalone'
- P30 L12-3: Define $A_F$
- P31 L1: 'heigh' → 'height'
- P32 L24 and L32: Check reference formatting

We thank the reviewer for the careful inspection of the manuscript and have adopted it accordingly.

---

## Author Comment (AC2) · 16 Jan 2020

**Reply to reviewer's comments**

Typographical convention: Black -> reviewer's comment Blue -> author's comment Red -> modification to the manuscript Underlined -> text added to the original version Strikethrough -> text removed from the original version

The line numbers refer to the original version of the manuscript.

**Reviewer 2 (RC2)**

This paper describes and evaluates COSMO-BEP-Tree v1.0: a coupled urban climate model with explicit representation of street trees. The authors assess the model during a heatwave event with observations from flux towers, surface stations and satellites. The authors have presented a very strong study which is thorough and convincing. The issue of integrating the effects of street trees in urban meteorological models is important because increasing tree cover is "go-to" response of urban planners who wish to reduce urban heat impacts. As the manuscript shows, interacting processes are complex and outcomes not always obvious. The evaluation is clear and thorough, and the technical achievements will be beneficial for future urban studies.

The lack of open availability of the source code is disappointing for a study presented in GMD. The process for accessing code outlined in the study is not timely, and includes mailing physical documents and/or negotiation. I have therefore not reviewed the code. The authors should reconsider publishing the code openly on a persistent public archive. Detailed comments follow.

**We are happy that the reviewer appreciated the paper.**

We regret that the source code was not easily accessible. Unfortunately, we have to conform to the CLM-Community licence agreement that only allows sharing the code with registered members. As such, we cannot publish the code on a public archive and we don't have control on the registration process.

However, we agree with the reviewer that the registration procedure must be simplified and made timely. We will make this point to the CLM-community.

Pg 2 ln 12: "Tree transpiration reduces the surface temperature of the foliage by converting part of the solar radiation to latent heat" not only solar radiation.

We thank the reviewer for pointing out the inaccurate description of the energy exchanged during transpiration.

We edited the sentence as follows:

**Pg 2 In 12: "In terms of heat interactions, Through tree transpiration the, leaf surface temperature is reduceds resulting in the extraction of sensible heat from the air of the foliage by converting the solar radiation to latent heat (Green, 1993)."**

Pg 2 In 15: "Modelling studies on the cooling potential of street trees have almost exclusively been performed at the scale of individual street canyons to single neighbourhood...Only very few studies investigate the city-wide impact of street trees, mainly due to the limited availability of models able to represent street trees at the scale of the city..." It would be useful here to acknowledge the various mesoscale urban meteorological models which have incorporated vegetation as low-height gardens within the street canyons (e.g. Thatcher and Hurley, 2012; Lemonsu et al, 2012; Wang et al 2013). Although not "street-trees" (as they do not provide shading on walls or reduce canyon sky view), they do shade the ground and interact directly with canyon radiation/turbulent fluxes and momentum budgets, alter the Bowen ratio and reduce canyon surface temperatures. This will also give an opportunity to be clearer about the authors definition of "street trees" when the concept is introduced, as readers may assume models within-canyon low vegetation, or even external vegetation tiles, are doing the same thing. The statement "very few studies investigate the city-wide impact of street trees" could then be more carefully stated, as there have been many studies at city or larger scales which have assessed the impact of urban vegetation, but most have used schemes with low vegetation or used a tiled approach, and hence missed important shading and sky view effect which is the strength of the current study.

\* Thatcher M and Hurley P 2012 Simulating Australian urban climate in a mesoscale atmospheric numerical model Boundary-Layer Meteorol 142 149–75

\* Lemonsu A, Masson V, Shashua-Bar L, Erell E and Pearlmutter D 2012 Inclusion of vegetation in the Town Energy Balance model for modelling urban green areas Geosci. Model Dev. 5 1377–93
\* Wang Z-H, Bou-Zeid E and Smith J A 2013 A coupled energy transport and hydro-logical model for urban canopies evaluated using a wireless sensor network Quarterly Journal of the Royal Meteorological Society 139 1643–57

We thank the reviewer for these excellent suggestions. We re-worded the section in order to acknowledge studies using the tile approach and those explicitly representing low vegetation.

Pg 2 In 15: "Modelling studies on the cooling potential of street trees have almost exclusively been performed at the scale of individual street canyons to a single neighbourhood (e.g. Gromke et al., 2015; Ng et al., 2012). Only very few studies investigated the city-wide impact of street trees, mainly due to the limited availability of models able to represent street trees at the scale of the city or at even larger scales. In fact, the vast majority of meso-scale urban climate models only represent trees vegetation outside the street canyon, neglecting important effects such as the shading effect of trees on the canyon's surfaces."

Pg 2 In 15: "The climatic impact of urban vegetation has been investigated in numerous previous studies from the scale of the single street canyon to that of the entire urban region (e.g. Gromke et al., 2015; Ng et al., 2012; De Munck et al., 2018). However, studies on entire urban regions primarily focused on low vegetation, representing low height gardens and green roofs (e.g. Wang et al., 2013; De Munck et al., 2018). Street trees, instead, have generally only been represented in a separate natural tile (e.g. Schubert and Grossman-Clarke, 2013; Li and Norford, 2016). This approach precluded considering any interactions between trees and urban surfaces in a street canyon, such as shading and sheltering effects (Krayenhoff et al. 2014, 2015). Other studies employed somewhat more sophisticated methods but still neglect some of the critical interactions between trees, canyon surface and airflow (e.g. Thatcher and Hurley, 2012)."

New references:

Schubert, Sebastian, and Susanne Grossman-Clarke. "The Influence of green areas and roof albedos on air temperatures during Extreme Heat Events in Berlin, Germany." Meteorologische Zeitschrift 22.2 (2013): 131-143. https://doi.org/10.1127/0941-2948/2013/0393

De Munck, C., et al. "Evaluating the impacts of greening scenarios on thermal comfort and energy and water consumptions for adapting Paris city to climate change." Urban Climate 23 (2018): 260-286. https://doi.org/10.1016/j.uclim.2017.01.003

Li, Xian-Xiang, and Leslie K. Norford. "Evaluation of cool roof and vegetations in mitigating urban heat island in a tropical city, Singapore." Urban Climate 16 (2016): 59-74. https://doi.org/10.1016/j.uclim.2015.12.002

Krayenhoff, E. S., et al. "A multi-layer radiation model for urban neighbourhoods with trees." Boundary-layer meteorology 151.1 (2014): 139-178. https://doi.org/10.1007/s10546-013-9883-1

Krayenhoff, E. S., et al. "Parametrization of drag and turbulence for urban neighbourhoods with trees." Boundary-layer meteorology 156.2 (2015): 157-189. https://doi.org/10.1007/s10546-015-0028-6

Thatcher, Marcus, and Peter Hurley. "Simulating Australian urban climate in a mesoscale atmospheric numerical model." Boundary-layer meteorology 142.1 (2012): 149-175. https://doi.org/10.1007/s10546-011-9663-8

pg 6 ln 4: Why are the intensity terms ( $\Delta V_i$  and  $r_i$ ) dimensionless? Shouldn't they have units?

We apologize that the units were missing. Both terms indeed have units of  $[W m^{-2}]$ . The sentence will be changed as follows:

pg 6 ln 4: "where  $\Delta V_i$  is the reduction in intensity of ray i due to the tree foliage [W m-2], ri is the initial intensity of the ray i [W m-2], ... "

pg 8 ln 13 "street trees do not interact with soil moisture content as represented by COSMOS's land surface scheme" More explanation as to why this was the case –technical constraints? Or was it

assumed that all soil is constantly irrigated in urban areas? You may wish to mention the drawback of this assumption (i.e. over estimating latent heat) at this time. Otherwise, do the underlying LSM soil and urban scheme exchange any fluxes? How? Figure 1 could be used here to better explain how fluxes of the urban scheme are coupled to the atmosphere and LSM.

This point was also raised by the other reviewer. We added further discussion on the lack of interaction with soil moisture.

Further motivation is provided in the section "Methodology"

pg8 In 12: "Although moisture exchange between street trees and the atmosphere is implemented, street trees do not interact with soil moisture content as represented by COSMO's land surface scheme. In other words, a mechanistic interaction between soil moisture and the transpiration of street trees is not included, assuming that the transpiration is never limited by soil water availability. A careful representation of soil moisture in the urban tile would have required a new urban hydrology scheme, which was beyond the scope of the study. The missing interaction with soil moisture may reduce the model ability to represent variations in transpiration during periods with large changes in soil moisture (Konarska et al., 2016; Asawa et al., 2017). Nevertheless, street trees are less sensitive to variations in soil moisture than short vegetation, thanks to their deeper root system (Chen et al., 2011; Asawa et al., 2017)."

**References:**

Chen, Lixin, et al. "Biophysical control of whole tree transpiration under an urban environment in Northern China." *Journal of Hydrology* 402.3-4 (2011): 388-400. https://doi.org/10.1016/j.jhydrol.2011.03.034

Asawa, Takashi, Tomoki Kiyono, and Akira Hoyano. "Continuous measurement of whole-tree water balance for studying urban tree transpiration." *Hydrological Processes* 31.17 (2017): 3056-3068. https://doi.org/10.1002/hyp.11244

Konarska, Janina, et al. "Transpiration of urban trees and its cooling effect in a high latitude city." International journal of biometeorology 60.1 (2016): 159-172. https://doi.org/10.1007/s00484-015-1014-x

pg 10 ln 6: For the last terms of eq 13, why is Ts associated with fnat, and Tg be with furb? I don't understand whether this temperature averaging approach is valid. I can see the equation is taken from Schubert (2013), but it is not explained there either. For example, take the extreme position where fnat=0, furb=1, then the equation simplifies to: T2m = Ts + r (T1 - Tg) Why is Tg part of the T2m temperature if there is no fnat? Also, how is the urban surface temperature here defined? Does it, for example, include the roof facets? I was pleased to see the clear definition of surface temperature for the evaluation of the satellite observations (pg 17) which was based on the facets that the satellite sees (roofs streets and natural surfaces rather than walls). But it's not clear that definition is appropriate for 2m temperature out of sight of roofs, and within sight of walls. If the same satellite-based definition is used for the T2m calculation, authors should note that's likely to lead to underestimation of the cooling benefit of street trees for temperatures within the canyon. We apologize that the terms were mixed up in the second part of the equation.  $f_{nat}$  should always go with  $T_g$  and  $f_{urb}$  with  $T_s$ . Additionally, the definition of  $T_s$  was not consistent with its use later in the manuscript (where it is referred to as  $T_{str}$ ).

We corrected the formula and the following explanation.

P10, L6 "T2m = (fnat Tg + furb  $\mp_{\underline{s}} \underline{T}_{\underline{str}}$ ) + r (T1 - (fnat  $\#_{\underline{s}} \underline{T}_{\underline{g}} + f_{urb} \#_{\underline{g}} \underline{T}_{\underline{str}}$ )),

where  $T_g$  is the surface temperature of the natural tile,  $T_{str}$  is the street surface temperature of the urban tile, ... "

pg 11 ln 3: It would be useful to include information here about the average fraction of green cover in Basel, for easy comparison with other urban studies.

We thank the reviewer for the suggestion. However, we think that the average fraction of green cover (as normally defined in urban climate studies) is not the best measure, since it includes all urban vegetation (e.g. parks), not only street trees, which is the focus of the present study. Instead, we calculated the average canopy cover from street trees only.

Additionally, we included an additional figure that shows the variability of the canopy cover from street trees in the different districts (Figure R1). The figure is added to the Supplement.

pg 11 ln 5: "The average canopy cover from street trees is about 20% when considering the entire urban area, but it varies from 6 to 31% in the different districts (see Supplement)."

Figure R1 - Canopy cover from street trees in the administrative districts of Basel-Stadt. The building geometries are shown in the background.

p 11 ln 23: There is discussion of atmospheric initial conditions and on soil properties but it is not clear how soil moisture was initialised. Soil moisture has significant impact on the intensity of

heatwaves (e.g. see Wang et al 2019), and soil moisture has memory much longer than the 5-day spin up, so this variable needs to be carefully initialised.

\* Wang P, Zhang Q, Yang Y and Tang J 2019 The Sensitivity to Initial Soil Moisture for Three Severe Cases of Heat Waves Over Eastern China Front. Environ. Sci. 7

Soil moisture is indeed a critical quantity in these simulations. Soil moisture (in the non-urban tiles) is computed prognostically for 7 soil layers by the land-surface module TERRA of the COSMO model. Initial soil conditions for our simulations were taken from the operational COSMO-2 analyses of MeteoSwiss. Since the COSMO-2 model (covering the Alpine domain) is nested into the COSMO-7 model (covering Europe), which itself is nested into ECWMF's IFS model, the initial soil moisture field ultimately goes back to the soil moisture analyses of ECMWF's operational IFS model.

We edited the sentence regarding initial and boundary conditions by specifically mentioning that they include soil moisture.

P 11 ln 21: "Soil moisture, a key variable for accurate representation of heatwaves (Wang et al., 2019), is also initialised with the COSMO-2 analyses, which trace back to the soil moisture analyses of ECMWF's operational IFS model (De Rosnay et al, 2013)."

New references:

Wang, Pinya, et al. "The sensitivity to initial soil moisture for three severe cases of heat waves over Eastern China." Frontiers in Environmental Science 7 (2019): 18. https://doi.org/10.3389/fenvs.2019.00018

De Rosnay, Patricia, et al. "A simplified Extended Kalman Filter for the global operational soil moisture analysis at ECMWF." Quarterly Journal of the Royal Meteorological Society 139.674 (2013): 1199-1213. https://doi.org/10.1002/qj.2023

pg 13 Table 1, please provide explanation for name acronyms here for easier reference.

We extended Table 1 in order to ease the reference.

| Name | $L_{De}$ | g s | Description                    |
|------|----------|------------|--------------------------------|
| STD  | Std      | Std        | Current conditions             |
| LA0  | 0        | Std        | No street trees                |
| LA+  | +50%     | Std        | Increased leaf area density    |
| LA-  | -50%     | Std        | Decreased leaf area density    |
| SC+  | Std      | +50%       | Increased stomatal conductance |
| SC-  | Std      | -50%       | Decreased stomatal conductance |

pg 18 ln 4. I didn't find the attribution of the night-time overestimation of air temperature to the inability of COSMO to represent nocturnal stable boundary layer conditions convincing. If it were an

issue with COSMO failing to represent a stable boundary layer, then it should occur in the non-urban site (BLER) night-time cooling rate. However, the supplementary Figure 1 indicates the rate of cooling matches observations well (al-though with a positive bias throughout), which to me indicates the evolution of stability conditions are probably reasonable at the rural site. It therefore seems the overestimation of night-time urban air temperatures could just as easily be from issues within the urban scheme configuration rather than the atmospheric model or stability issues. The supplementary Figure 2 indicates the bulk albedo may be low, and the material properties of facets listed in Table 2 store a lot of heat. I understand those values are based on previous studies, but those values were derived through optimising WRF SLUCM and so are not necessarily realistic urban parameters, but could simply be accounting for deficiencies in that model. Therefore, the parameters won't necessarily be appropriate for BEP-TREE at Basel. I'm not asking for the simulations to be redone, but the authors shouldn't be so quick to simply attribute night-time errors to the atmospheric model when urban parameters may be the issue (low albedo, high heat storage).

We thank the reviewer for pointing out this important aspect.

By re-examining the figures, we agree with the reviewer that the attribution of the night-time overestimation of air temperature at the BKLI tower site (urban site) cannot be explained by the model inability to representing the stable boundary layer.

We agree that the bias may be due to the choice of material properties, especially the too low albedo (as substantiated by the radiation fluxes – Supplement, Figure 2). However, we remark that some aspects are still inconsistent with the observations, such as the fact that a clear bias was not present at the 2-m stations.

In view of these arguments, we edited the paragraph as follows:

pg 18 ln 1: "The model simulates the evolution of air temperature very well during the evaluation period (Figure 5a) although a slight overestimation with a MBE of 0.51 K and a systematic RMSE of 0.51 K was found. The overestimation occurs mostly during night-time and it seems to be related to an underestimation of the bulk albedo (Supplement) which can be traced back to the choice of material properties (Table 2). A too low albedo may have produced an excess in heat storage with consequently larger sensible heat release at night. Nevertheless, a night-time overestimation of temperature was not found at the near-surface sites, indicating that more analyses are needed to better understand eventual issues with the choice of material properties.", an issue that was observed also in other urban climate studies with COSMO (Schubert and Grossman-Clarke, 2014; Wouters et al., 2016) and other models (e.g. Lee et al., 2011). The night-time overestimation of air temperature is attributed to the inability of COSMO to represent very stable boundary layer conditions (Cerenzia, 2017)."

Additionally, we added a line about this issue in the "Future Work" section.

Pg 27 In 16: "Even if generally comparable with other models, the performance of COSMO-BEP-Tree still has room for improvements. The model evaluation against urban flux tower measurements revealed a systematic underestimation of specific humidity (q). The source of this bias needs to be

better investigated by, for instance, (a) analysing the soil moisture content and (b) evaluating q at the model boundaries provided by another model. The model evaluation reveals an overestimation of night-time air temperature above the urban canopy layer, which can be due to the use of default values of material properties. ..."

pg 20 ln 27 "are" -> area

Thank you for pointing out the typo.

pg 20 ln 27: "... and leaf area index from street trees ..."

pg 26 "Future Work" It was noted earlier that BEP-TREE does not interact with LSM soil moisture - this is a major limitation which should be discussed here.

We extended the section "Future Work" accordingly.

pg26, In 18: "In order to improve the representation of the transpiration from street trees and therefore the modelling of the associated latent heat fluxes, a mechanistic stomata model (e.g. Damour et al. (2010)) will have to be implemented. Furthermore, to properly represent soil water scarcity during extended drought periods, an urban hydrology model would have to be implemented (e.g. Järvi 2011, Yang 2015, Stavropulos-Laffaille 2018)."

**References:**

Järvi, Leena, C. S. B. Grimmond, and Andreas Christen. "The surface urban energy and water balance scheme (SUEWS): Evaluation in Los Angeles and Vancouver." *Journal of hydrology* 411.3-4 (2011): 219-237. https://doi.org/10.1016/j.jhydrol.2011.10.001

Yang, Jiachuan, et al. "Enhancing hydrologic modelling in the coupled weather research and forecasting–urban modelling system." *Boundary-Layer Meteorology* 155.1 (2015): 87-109. https://doi.org/10.1007/s10546-014-9991-6

Stavropulos-Laffaille, Xenia, et al. "Improvements to the hydrological processes of the Town Energy Balance model (TEB-Veg, SURFEX v7. 3) for urban modelling and impact assessment." Geoscientific Model Development 11.10 (2018): 4175-4194. https://doi.org/10.5194/gmd-11-4175-2018

pg 27 ln 21 "attributed to the inability of the model to reproduce a very stable atmospheric conditions" again, if this argument is used it should be better supported with information from the current study. For example, was the minimum value of turbulent diffusion coefficients (K\_i,min) set at 1 m2/s (per Buzzi 2011) in the current experiments? What were the stability conditions at the rural site? Did an inversion form? Why is the rate of cooling at night at the rural site seemingly correct if a stability is incorrectly simulated? Do the observed wind speeds support the conclusion that very stable atmospheric conditions should have formed on those nights? Is there other observational

support for the stability or non-stability of rural sites in the area (non-urbanflux towers, boundary layer measurements etc)?

**We thank the reviewer for pointing out this aspect.**

We included an additional figure (Figure R2, added to the Supplement) that supports our argument regarding the formation of stable boundary layer conditions. By comparing the air temperatures at the rural reference site BLER (2 m a.g.) and at neighbouring St. Chrischona Tower (STCT, 230 m above ground, 3.4 km away from BLER), we can see that stable boundary layer conditions (characterised by a strong temperature inversion) were present during several nights.

Figure R2 - Air temperature evolution at the sites of St. Chrischona (STCT, 230 m a.g., lat=47.571767, lon=7.687094) and BLER (2 m a.g.). Solid and dashed lines indicate observations (Obs) and model simulation (Mod), respectively.

Additionally, we can further see that the night-time air temperature at the rural site of BLER is overestimated, especially during nights with stable boundary layer conditions. By contrast, the model compares very well against the observation at the tower.

However, we agree with the reviewer that a misrepresentation of the stable boundary layer conditions in COSMO cannot fully explain the bias. This is especially clear in the central part of the period, where a positive bias is present throughout the day.

We further examined the data and found a large underestimation of albedo at the rural site (see Figure R3 below, added to the Supplement). For completeness, we included a new figure (in the Supplement) with all the components of the radiation balance at the rural site BLER.

Figure R3 - Average diurnal profile of observed (Obs) and modelled (STD, LA0) albedo. Albedo is calculated as the ratio between upward and downward short-wave radiation. The site BKLI is charachterized by an urban fraction (furb) of 0.79.

This large underestimation has been already found in another urban study with COSMO-CLM (Schubert et al., 2014) and it is attributed to a misrepresentation of albedo in the land-surface model of COSMO-CLM.

By default in COSMO-CLM, albedo values are determined as a function of soil type, soil moisture and plant fraction. However, there is no distinction between vegetation type and a constant background albedo value for vegetation of 0.15 is applied (Tölle et al., 2018).

For the configurations at the BLER site, the model estimates a daily average value of about 0.15. This value is somewhat smaller then what expected for the land cover at the site, where agricultural fields with crops are present (albedo = 0.18-0.25, Oke et al., 2018). It is not clear, however, whether this bias is a specific problem at the BLER site or a more general issue over the rural areas that surround the city.

Based on this analysis, we recommend that future studies consider the use of more advance method for albedo estimation, as developed and tested in recent studies with COSMO-CLM (e.g. Tölle et al., 2018).

In conclusion, we think that both aspects (inability to represent very stable boundary layer conditions by COSMO and underestimation of albedo) may play a role in determining the air temperature bias at the rural reference site BLER.

We reconsidered our analysis and made changes to the manuscript as follows:

pg 20 ln 23 "This is probably determined by the combined effect of underestimation in surface albedo and misrepresentation of atmospheric stability (Supplement). A similar underestimation of albedo at rural sites has been already found by Schubert and Grossman-Clarke (2014). The underestimation is attributed to the default albedo scheme of COSMO-CLM, which fails to represent different vegetation types. It is unclear, however, whether this bias is due to specific conditions at the BLER site (grassland) or due to a general misrepresentation in the land-surface model of COSMO- CLM. The misrepresentation of stable boundary layer conditions is a well-known issue of COSMO (Buzzi et al., 2011; Cerenzia, 2017) and has already been reported in previous studies (Schubert and Grossman-Clarke, 2014; Mussetti et al., 2019)."

pg 27, In 19-23 "The model evaluation at 2-m reveals an overestimation of the air temperature at the rural sites especially at night. Beside previous studies where this bias was attributed only to the misrepresentation of stable boundary layer conditions (Mussetti et al., 2019), the present study provides new evidences that point out to an underestimation of albedo over rural areas. Future studies need to address this issue that partially limits the model ability to represent the urban heat island effect. Potential solutions include the use of more advanced representations of albedo, already available as options in COSMO-CLM, with explicit consideration of vegetation type or satellite-based albedo values (Tölle et al., 2018). This issue has been already found in other studies (e.g. Schubert and Grossman-Clarke, 2014) and is attributed to the inability of the model to representation of stable atmospheric conditions (Buzzi et al., 2011).-In respect to the representation of stable boundary layer conditions, more recent versions of the COSMO model (from version 5.4a onward) promise a better performance thanks to a revised turbulence scheme."

**References:**

Schubert, Sebastian, and S. Grossman-Clarke. "Evaluation of the coupled COSMO-CLM/DCEP model with observations from BUBBLE." Quarterly Journal of the Royal Meteorological Society 140.685 (2014): 2465-2483. https://doi.org/10.1002/qj.2311

Tölle MH, Breil M, Radtke K and Panitz H-J (2018) Sensitivity of European Temperature to Albedo Parameterization in the Regional Climate Model COSMO-CLM Linked to Extreme Land Use Changes. Front. Environ. Sci. 6:123. https://doi.org/10.3389/fenvs.2018.00123

Finally, I found myself referring to the rural figure often throughout the manuscript, it may be better placed in the main body near figure 6.

We added the daily evolution of  $T_{2m}$  at the rural reference site BLER to Figure 6.

pg 21, caption Figure 6: "Figure 6. Comparison between period averaged daily profiles of observed (Obs) and model-simulated 2-m air temperature at the sites BFEL, BSJO, SLFR-and BBIN and BLER during the selected period (discarding the first 5 days as spin-up). Black dots indicate the observations. Blue lines and red lines indicate the results from the STD and LAO model runs, respectively. The range of variability within the selected period is shown as shaded area for Obs (grey) and STD (light blue)."

pg 28 ln 16 "additionally explained by an underestimation of the night-time temperature" I believe the night-time temperature is overestimated at the BKLI site.

We thank the reviewer for pointing out the typo.

pg 28 ln 16: "At the most urban site, it is additionally explained by an overunderestimation of the night-time temperature."

---

## Author Response (AR2)

**Reply to editor's comments**

**Dear Editor,**

We are glad to hear that we have successfully implemented the suggestions from the reviewers.

Please find below a point-by-point response to your technical comments, a list of relevant changes made to the manuscript, and a marked-up manuscript version (including the supplement).

Thank you and best regards, Gianluca Mussetti on behalf of the co-authors

**Editor's comments**

**Dear authors,**

You have done a good work in responding the referee comments. I have only a few technical comments for the manuscript that should be addressed before it can be accepted for final publication to GMD:

- The equations, figures and tables in supplements should be numbered as (S1), Fig. S5 or Table S6 etc. Sections on the other hand should be numbered as S3, S3.1, and S3.1.1.

We amended the numbering of figures and tables in the supplement.

- And furthermore the figures/tables should be referred from the main manuscript based on the figure and table numbers. For example on P6, L6 where (Supplement) should be (Fig. S1, Table S1) but also elsewhere in the manuscript

We included explicit reference to figures/tables in the supplement. Additionally, we re-ordered the figures in the supplement according to the order in which they appear in the main manuscript.

- Table 1-3: Horizontal lines should appear above and below the table. Now in Table 3 there is only the below line. Later tables are OK.

We added (top and bottom) horizontal lines as requested (Table 1, Table 2, Table 3).

-Figure 8: Elsewhere local time is used instead of LT. Please, systematize.

We changed all the occurrences of "LT" to "local time" (P24 Caption Table 7, P24 L2 and P24 Caption Figure 8).

-P28, L28: references extend over the side

We do not understand the issue here.

Breaking the reference at the end of the page side should not be a problem as this is consistently done in papers published in GMD. Just taking the first two "Highlight articles" in GMD we found two examples: https://doi.org/10.5194/gmd-13-335-2020 (L3 of Introduction) and https://doi.org/10.5194/gmd-12-5113-2019 (L3 of Introduction).

-References with websites (urls) should be avoided as they might change in the future. In Code and Data Availability section both https://www.clm-community.eu/ should be removed. It is also tricky to use Dominik Brunner's email as emails are not permanent so rather say upon request etc or leave simply "The code of COSMO-BEP-Tree is made freely available to any user with a valid CLM-Community membership". Also the references to MODTRAN should be changed to those suggested in their website:
* * *
What reference(s) should be used for MODTRAN?

Recent MODTRAN publications include [Berk et al., 2014; 2015]. The 2014 paper should be used as a general MODTRAN reference, while the 2015 paper focuses on the MODTRAN line-by-line algorithm:

1. A. Berk, P. Conforti, R. Kennett, T. Perkins, F. Hawes, and J. van den Bosch, "MODTRAN6: a major upgrade of the MODTRAN radiative transfer code," Proc. SPIE 9088, Algorithms and Technologies for Multispectral, Hyperspectral, and Ultraspectral Imagery XX, 90880H (June 13, 2014); doi:10.1117/12.2050433.

2. Alexander Berk, Patrick Conforti, and Fred Hawes, "An accelerated line-by-line option for MODTRAN combining on-the-fly generation of line center absorption with 0.1 cm-1 bins and precomputed line tails," Proc. SPIE 9471, Algorithms and Technologies for Multispectral, Hyperspectral, and Ultraspectral Imagery XXI, 947217 (May 21, 2015); doi:10.1117/12.2177444

We edited the section "Code and Data Availability" and removed the references to websites and email addresses. Where possible, we included a DOI reference instead.

P30: "Code and data availability. COSMO-CLM is freely available for scientific usage to the members of the CLM-Community. The procedure to become a member is available on the community website-(https://www.clm-community.eu/) or by contacting the coordination office (clmcoordination@dwd.de). The code of COSMO-BEP-Tree is made freely available to any user with a valid CLM-Community membership (contact Dominik Brunner dominik.brunner@empa.ch). All research data and scripts to produce figures and analysis are archived on Zenodo (Mussetti, 2019a). UCPgenerator v1.0 can be downloaded from the GitHub repository (Mussetti, 2019b). The LUCY model, used to calculate the anthropogenic heat flux, can be downloaded from Grimmond et al. (2018). BEP-Tree as a standalone model can be obtained by contacting Scott Krayenhoff (skrayenh@gmail.com) is available under request to Krayenhoff et al. (2020). MODTRAN is a commercial software ÷ http://modtran.spectral.com/ (accessed 26 08 2019) (Berk et al., 2014)."

The reference paper of Berk et al., 2014 is also added to the manuscript when the reference to MODTRAN is made (P17, L12).

**Additional changes/corrections by the authors**

P18, L16 -> cross link added to Table 2

P33, L24-25 -> A line was added to acknowledge the work of the reviewers

**COSMO-BEP-Tree v1.0: a coupled urban climate model with explicit representation of street trees**

Gianluca Mussetti1,2,3, Dominik Brunner1, Stephan Henne1, Jonas Allegrini2,3, E. Scott Krayenhoff4, Sebastian Schubert5, Christian Feigenwinter6, Roland Vogt6, Andreas Wicki6, and Jan Carmeliet3 1Laboratory 
[revised manuscript text omitted]
_{\rm urb}^{\downarrow} = \frac{(K_{\rm urb}^{\uparrow} - \alpha_{\rm urb}^{\downarrow} K^{\downarrow})}{K^{\downarrow}},$$
 (7)

with  $K_{\rm urb}^{\uparrow}$  being the shortwave radiation reflected from all street canyon elements into the sky.

The radiative surface temperature is calculated as:

$$T_{\rm rad} = \left(\frac{L_{\rm urb}^{\uparrow} - (1 - \epsilon_{\rm urb})L^{\downarrow}}{\sigma \epsilon_{\rm urb}}\right)^{1/4},\tag{8}$$

where  $L_{\text{urb}}^{\uparrow}$  is the sum of the emitted and reflected long-wave radiation from all street canyon elements into the sky,  $\epsilon_{\text{urb}}$  is the 15 bulk emissivity of the urban surface,  $L^{\downarrow}$  is the incoming long-wave radiation and  $\sigma$  is the Stephan-Boltzmann constant.  $\epsilon_{\text{urb}}$  is calculated as the mean emissivity of the street canyon elements.

The bulk radiation parameters from the urban tile  $(_{urb})$  computed by BEP-Tree are combined with the ones computed by COSMO for the natural  $(_{nat})$  tile as:

$$\alpha^{\downarrow} = f_{\rm urb} \, \alpha^{\downarrow}_{\rm urb} + f_{\rm nat} \, \alpha^{\downarrow}_{\rm nat}, \tag{9}$$

20
$$\alpha^{\downarrow} = f_{urb} \alpha^{\downarrow}_{urb} + f_{nat} \alpha^{\downarrow}_{nat},$$

$$\epsilon = f_{\rm urb} \,\epsilon_{\rm urb} + f_{\rm nat} \,\epsilon_{\rm nat},\tag{11}$$

(10)

$$T_{\rm rad} = \left(\frac{\left(f_{\rm urb}\,\epsilon_{\rm urb}\,T_{\rm urb}^4 + f_{\rm nat}\,\epsilon_{\rm nat}\,T_{\rm nat}^4\right)}{\epsilon}\right)^{1/4},\tag{12}$$

[revised manuscript text omitted]